# A Comprehensive Survey on Trustworthiness in Reasoning with Large Language Models

## Abstract

The development of Long-CoT reasoning has advanced LLM performance across various tasks, including language understanding, complex problem solving, and code generation. This paradigm enables models to generate intermediate reasoning steps, thereby improving both accuracy and interpretability. However, despite these advancements, a comprehensive understanding of how CoT-based reasoning affects the trustworthiness of language models remains underdeveloped. In this paper, we survey recent work on reasoning models and CoT techniques, focusing on five core dimensions of trustworthy reasoning: truthfulness, safety, robustness, fairness, and privacy. For each aspect, we provide a clear and structured overview of recent studies in chronological order, along with detailed analyses of their methodologies, findings, and limitations. Future research directions are also appended at the end for reference and discussion. Overall, while reasoning techniques hold promise for enhancing model trustworthiness through hallucination mitigation, harmful content detection, and robustness improvement, cutting-edge reasoning models themselves often suffer from comparable or even greater vulnerabilities in safety, robustness, and privacy. By synthesizing these insights, we hope this work serves as a valuable and timely resource for the AI safety community to stay informed on the latest progress in reasoning trustworthiness. A full list of related papers will be made public.

## 1 Introduction

With the advancement of large language models (LLMs), Chain-of-Thought (CoT) techniques have become an important way to improve model performance on various downstream tasks, especially in math and code generation. After the release of OpenAI's o1 series models as well as the DeepSeek-R1, developing reasoning models with system-2 thinking also attracted significant interest from researchers around the world, followed by innovations in reinforcement learning algorithms, training data generation, and adaptation methods for other tasks.

Despite these improvements, the trustworthiness of CoT techniques as well as reasoning models remains underexplored. Intuitively, it may be reasonable that the thinking capability could be generalized to the trustworthiness domain, resulting in a safer and more reliable model. However, recent works (Jiang et al., 2025b; Lu et al., 2025a; Ying et al., 2025b) did not support such an ideal hypothesis. Furthermore, prior surveys on LLM safety (Wang et al., 2025d; Dong et al., 2024b; Shi et al., 2024) provide little discussion of reasoning as a factor in model trustworthiness. This gap motivates the central question: **What does the reasoning capability bring to the language model trustworthiness?**

To answer this question, we propose the first comprehensive survey to thoroughly review recent advancements in trustworthy reasoning. We unfold our survey through five main components: truthfulness, safety, robustness, fairness, and privacy. In the truthfulness section, with a focus on model reliability, we include hallucination and reasoning faithfulness, encompassing hallucination detection and mitigation methods with CoT techniques, hallucination analysis in reasoning models, reasoning faithfulness measurement, faithfulness understanding, as well as methods to improve reasoning faithfulness. In the safety section, we aim to understand the harmlessness of the generation content, and mainly take vulnerability assessment, jailbreak,

alignment, and backdoor into consideration. For better readability, we specifically distinguish between jailbreak attacks targeting reasoning models and the use of reasoning techniques in attack and defense, forming different paragraphs to structure the literature. In the robustness section, we mainly focus on adversarial input noises that elicit false answers at inference time. The overthinking and underthinking problems are highlighted as a special case when language models are equipped with reasoning capability. After that, in the fairness section, we mainly cover the latest evaluations and methods for bias detection. As for the privacy section, we split the related works into model-related privacy and prompt-related privacy, with topics containing model unlearning, IP protection, watermarking, and privacy inference.

While existing surveys have explored reasoning techniques (Chen et al., 2025b; Xu et al., 2025a) and reasoning efficiency (Qu et al., 2025; Sui et al., 2025; Feng et al., 2025b), relatively little attention has been paid to the trustworthiness of reasoning in large language models. A related survey (Wang et al., 2025c) provided valuable discussions on safety-related aspects. In contrast, our work offers a more comprehensive perspective on trustworthiness. In general, we provide a clear taxonomy for model trustworthiness in reasoning, which includes both early CoT techniques and end-to-end reasoning models. Through our review of existing work, we suggest that reasoning techniques not only facilitate the development of more interpretable and trustworthy models but also introduce new vulnerabilities. As models acquire more advanced reasoning capabilities, the attack surface correspondingly expands, enabling more complex and targeted adversarial strategies. We hope that both the surveyed literature and our proposed taxonomy will serve as a timely reference for the AI safety community, supporting ongoing efforts to understand and improve the trustworthiness of reasoning in language models.

Table 1: List of Abbreviations and Acronyms

| Abbreviation | Full Term | Abbreviation | Full Term |
|---|---|---|---|
| AOC | Area Over Curve | MCTS | Monte-Carlo Tree Search |
| ASR | Attack Success Rate | MLLM | Multimodal Large Language Model |
| CNN | Convolutional Neural Network | MLRM | Multimodal Large Reasoning Model |
| CoT | Chain-of-Thought | ORM | Outcome Reward Model |
| DFS | Depth-First Search | PRM | Process Reward Model |
| DPO | Direct Preference Optimization | QA | Question-Answering |
| GRPO | Group Relative Policy Optimization | RL | Reinforcement Learning |
| ICL | In-Context Learning | RLHF | Reinforcement Learning from Human Feedback |
| KL | Kullback-Leibler Divergence | RLVR | Reinforcement Learning with Verifiable Reward |
| LAS | Leakage-Adjusted Simulatability | RAG | Retrieval-Augmented Generation |
| LLM | Large Language Model | SCM | Structural Causal Model |
| LRM | Large Reasoning Model | SoTA | State-of-the-Art |
| LoRA | Low-Rank Adapter | SFT | Supervised Fine-Tuning |
| MoE | Mixture-of-Experts | VR | Verifiable Reward |

## 2 Background

In this section, we provide an overview of fundamental concepts related to reasoning in language models, including discussions of the general definition of reasoning, an introduction to CoT as a widely adopted technique, and key considerations in model training that influence the reasoning abilities.

### 2.1 Large Language Model Reasoning

LLM reasoning is a novel paradigm that leverages the knowledge embedded within models like GPT-4 (Achiam et al., 2023), Claude (Anthropic, 2024), and DeepSeek-R1 (Guo et al., 2025b) to solve complex tasks—such as math, coding, and logical reasoning—by mimicking human cognitive processes. Typically, LLM reasoning involves generating both the final answer and the intermediate steps, often referred to as "thoughts", which guide the model from the question to the answer. Formally, given a prompt $x$ and context $C$, the reasoning of an LLM $\mathcal{M}$ can be represented as follows:

$$T, A = \mathcal{M}(x, C), \tag{1}$$

where $T$ refers to the intermediate reasoning process and $A$ is the answer. By enabling the AI system to generate interpretable reasoning steps alongside the solution, LLM reasoning not only solves complex tasks but also improves human understanding of the problem-solving process, thereby enhancing its utility and reliability. Currently, the two main paradigms for implementing large language model reasoning are CoT prompting and large reasoning model training.

## 2.2 Chain-of-Thought Prompting

CoT prompting (Wei et al., 2022; Kojima et al., 2022) is a prompt engineering technique designed to elicit a sequence of intermediate reasoning steps referred to as the thought, before providing the final answer. There are various methods for implementing CoT, with two of the most common being few-shot-CoT (Wei et al., 2022) and zero-shot-CoT (Kojima et al., 2022). As illustrated in Figure 1, few-shot-CoT mirrors the approach of few-shot in-context learning (ICL) (Brown et al., 2020), utilizing a small number of examples to guide the model in answering questions. Unlike traditional ICL, few-shot-CoT (Li et al., 2024e) not only shows the answer in the demonstrations, but also gives the specific reasoning steps before the answer. Therefore, the model will also give CoT before answering the question. While few-shot-CoT demonstrates strong performance on complex tasks such as math and symbolic reasoning, it requires human-annotated, task-specific examples with intricate reasoning paths, limiting its applicability. In contrast, zero-shot-CoT (Wei et al., 2022) offers a more flexible, task-agnostic method for eliciting CoT by simply adding the prefix "Let's think step by step" before generating the answer.

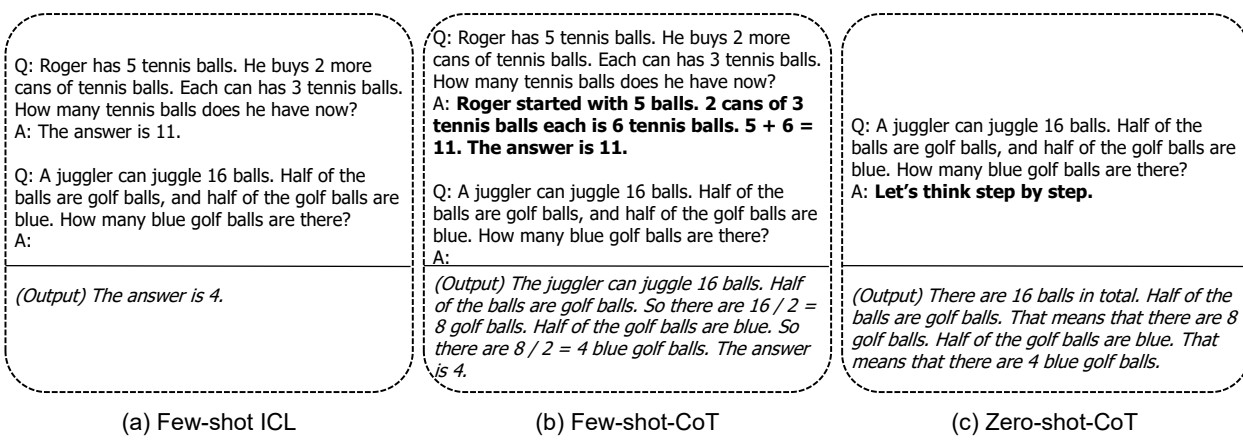

Figure 1: Illustration of typical CoT prompting. Few-shot-CoT uses several examples with the reasoning process to elicit CoT, and zero-shot-CoT uses a prefix prompt to induce the reasoning process.

## 2.3 Large Reasoning Models

Large reasoning models (LRMs), represented by OpenAI o1 (Jaech et al., 2024) and DeepSeek-R1 (Guo et al., 2025b), refer to a series of large language models that explicitly generate their thinking process before filling the final answers (Xu et al., 2025a). Instead of prompting models to "think step by step", reasoning models could automatically create the thinking process that mimics how humans analyze a problem.

### 2.3.1 Model Training

There are a few open-source trials to replicate the o1 series (Xu et al., 2025a), including OpenR (Wang et al., 2024b), o1-journey (Qin et al., 2024; Huang et al., 2024d; 2025f), and LLaMA-Berry (Zhang et al., 2024a). The key to the replication lies in distilling long CoT data, even if the source model has not been explicitly trained for reasoning. LLaMA-Berry (Zhang et al., 2024a) utilized Monte Carlo tree search (MCTS) (Browne et al., 2012) with a pairwise preference reward model to scale test-time compute, achieving a higher performance on multiple Math datasets such as GSM8k (Cobbe et al., 2021), MATH (Hendrycks et al., 2021),

GaoKao2023En (Liao et al., 2024), etc. O1-journey (Qin et al., 2024) utilized MCTS with a fine-grained reward model to construct long CoT data. After building the reasoning tree with each node annotated with a reward score indicating correctness, a traversal algorithm such as Depth-First Search (DFS) with constraints could be adopted to create a datapoint using an error-then-backtrack style. Supervised fine-tuning (SFT), followed by Direct Preference Optimization (DPO) (Rafailov et al., 2023), was then leveraged to train the reasoning model. OpenR (Wang et al., 2024b) introduced reinforcement learning with a process reward model to encourage reasoning capability. During training, the LLM policy was updated at each reasoning step using intermediate step-wise rewards from the reward model, optimized with either the proximal policy optimization (PPO) (Schulman et al., 2017) or the group relative policy optimization (GRPO) (Shao et al., 2024). Except for these tree searching methods, DeepSeek-R1 demonstrated the outstanding performance of pure reinforcement learning in boosting reasoning capability, utilizing distilled data from R1-Zero[1] to train the base model. One point worth noting is that, except for latent reasoning models (Geiping et al., 2025; Hao et al., 2024), there is no obvious difference between previous chat models and current reasoning models in terms of model structure. In fact, all these models are developed based on well-trained chat models such as DeepSeek-V3 (Liu et al., 2024a), Qwen2.5 (Qwen et al., 2024), and Llama-3 series (Dubey et al., 2024).

**PRM, ORM, and VR.** According to Uesato et al. (2022), current reward models could be divided into two types: process reward model (PRM) and outcome reward model (ORM), in which the former provides stepwise reward on each reasoning process, and the latter simply gives one score for the whole generation sequence. Instead of ORM (Zelikman et al., 2024), Lightman et al. (2023) proposed PRM to verify the thinking process step by step, and demonstrated its superior performance to ORM in providing more reliable step-wise reward. For inference-time scaling, these reward models could not only facilitate the tree search at inference time for better performance, but also help filter reasoning trajectories with higher quality for post-training. Before the release of DeepSeek-R1 (Guo et al., 2025b), the training of reward models is crucial for reasoning model development. Verifiable reward (VR) was first proposed by Lambert et al. (2024), which includes three types: correctness verification, verification via execution, and verifiable constraints (Mroueh, 2025). Different from reward models, here we define verifiable reward as "*the reward provided by a simple deterministic function instead of large models, which is objective, usually binary, and outcome-based*". DeepSeek-R1 demonstrates the effectiveness of VR, which is then regarded as a prevailing post-training method when combined with GRPO.

### 2.3.2 Multimodal LRM

Li et al. (2025d) summarized the development of multimodal large reasoning models (MLRMs) into three stages: "perception driven modular reasoning", "language-centric short reasoning", and "language-centric long reasoning". Like the development of unimodal large reasoning models, MLRMs also experienced the transformation from zero-shot or few-shot CoT prompting to long reasoning data post-training (Wang et al., 2025i). For example, Multimodal-CoT (Zhang et al., 2024g), VoT (Fei et al., 2024), and VIC (Zheng et al., 2024) are some of the early works that focused on the prompting to elicit model thinking. In terms of training, LLaVA-CoT (Xu et al., 2024a), Llamav-o1 (Thawakar et al., 2025), RedStar (Xu et al., 2025b), and Mulberry (Yao et al., 2024a) propose to empower multimodal large language models (MLLMs) with reasoning capabilities by fine-tuning base models. As stated in Section 2.3.1, multimodal CoT data generation is also crucial for model training, and the construction of the reasoning path includes distillation (Xu et al., 2024a; Thawakar et al., 2025; Zhang et al., 2024d; Dong et al., 2025b; Guo et al., 2024) or MCTS (Yao et al., 2024a; Sun et al., 2025b), which also resembles the way mentioned for text-domain CoT data generation.

As for model training, pure GRPO and SFT followed by GRPO become the prevailing method for reasoning model development (Wang et al., 2025i), which may be attributed to the outstanding performance of RL demonstrated by DeepSeek-R1.

---

[1]The model for long CoT data synthesis underwent preliminary supervised fine-tuning (cold start). Therefore, it is slightly different from the released R1-Zero model.

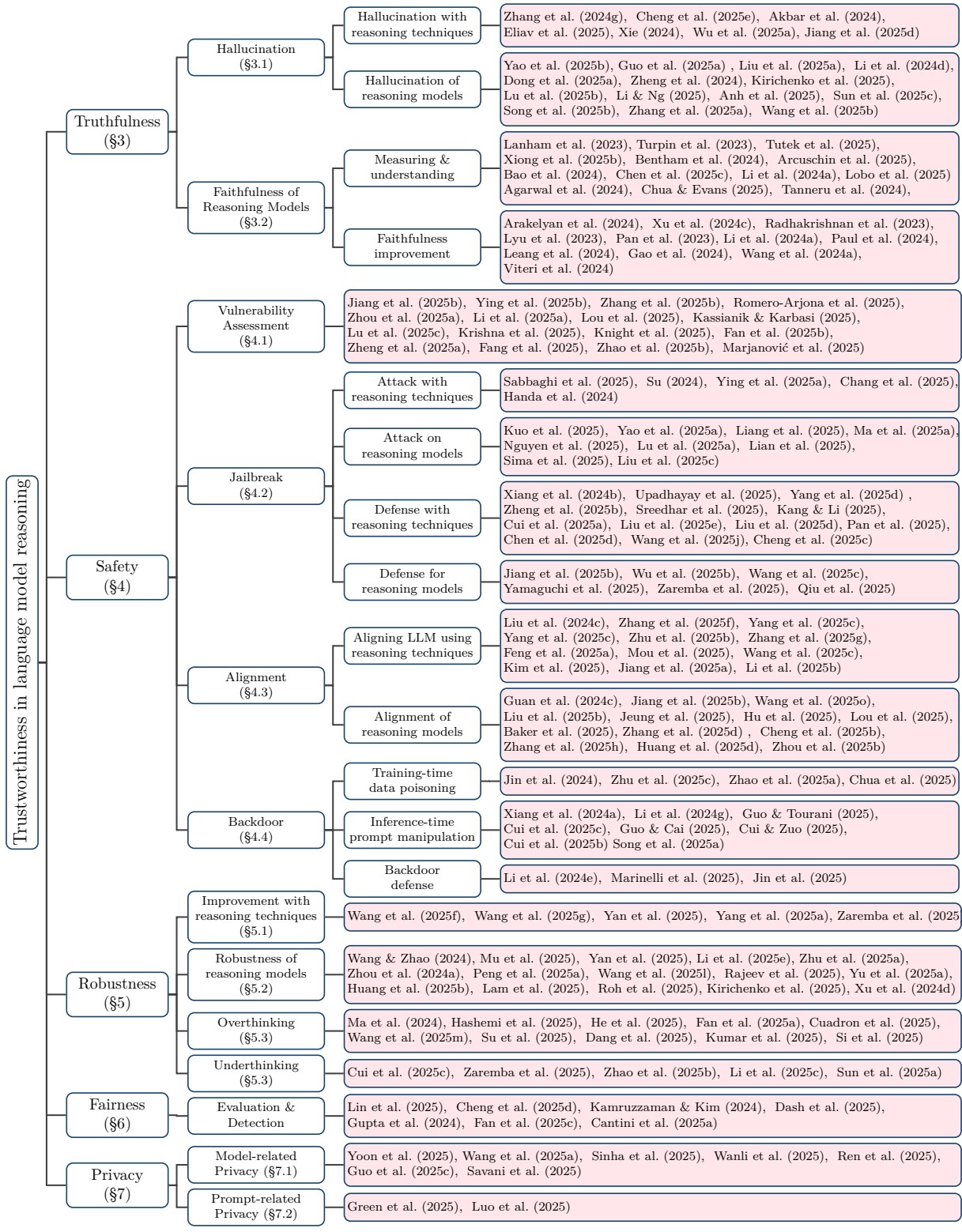

Figure 2: Taxonomy of trustworthiness in reasoning with large language models.

# 3 Truthfulness

Truthfulness in the LLMs refers to how an AI system accurately represents information, facts, and results (Huang et al., 2024c). This fundamental dimension of truthfulness focuses on the model's ability to provide factually correct and reliable information without generating misleading or false content. In this section, we discuss the new challenges brought by the reasoning techniques, including two aspects: hallucination and faithfulness.

## 3.1 Hallucination

*Hallucination in LLMs refers to instances where models generate responses that appear coherent and plausible but are inconsistent with the input, context, or factual information* (Huang et al., 2025c; Rawte et al., 2023). The emergence of reasoning models introduces new risks and challenges in managing hallucinations. First, reasoning models often generate responses that are more structured, logically coherent, and superficially persuasive, making them appear more reliable. As a result, hallucinated content from these models can appear more credible, making it harder for users to detect inaccuracies and increasing the risk of spreading misinformation (Sun et al., 2025c), especially in high-stakes fields such as healthcare, law, or education. On the other hand, the CoT reasoning generated by models can also contain hallucinations (Li & Ng, 2025). Compared to traditional LLMs, the hallucinations in reasoning models have not been as thoroughly evaluated. Moreover, the powerful reasoning capabilities of these models can be leveraged to detect or mitigate hallucinations in certain complex tasks (Cheng et al., 2025e; Eliav et al., 2025).

### 3.1.1 Hallucination with Reasoning Techniques

In this section, we explore how reasoning techniques can be leveraged to detect and mitigate hallucinations in LLMs. CoT prompting has shown remarkable success in addressing complex tasks (Kojima et al., 2022; Wei et al., 2022) and reducing hallucinations (Cheng et al., 2025a). To further enhance model reasoning capabilities, several techniques have been proposed, such as test-time scaling (Snell et al., 2024), self-consistency (Wang et al., 2023), etc. One such approach, HaluSearch (Cheng et al., 2025e), employed a tree search-based algorithm coupled with a switch model to determine when to engage in more deliberate, "slow thinking" processes. In contrast to hallucination mitigation, HalluMeasure (Akbar et al., 2024) focused on fine-grained hallucination measurement, using CoT prompting. Specifically, it decomposed model responses into a series of claims and applies CoT techniques to detect hallucinations at the claim level. Similarly, CLATTER (Eliav et al., 2025) adopted a multi-step reasoning process for hallucination detection, consisting of decomposition, attribution, entailment, and aggregation. Moreover, Xie (2024) observed that the order in which reasoning steps are applied can influence hallucination occurrence. As such, they propose Reflexive Prompting, which combines "answer-first" and "logic-first" reasoning strategies to improve model accuracy. Beyond text-based tasks, Zhang et al. (2024g) extended CoT to multimodal settings, proposing a method to mitigate visual hallucinations. Their approach involves generating a rationale that is used to update the language input, which is then combined with the original visual input to produce the final answer. Furthermore, Wu et al. (2025a) introduced Grounded Chain-of-Thought (GCoT), a technique in which the model gradually grounds visual cues before generating answers. This step-by-step process helps mitigate visual hallucinations by enhancing the model's understanding of the input. In addition, in the context of medical report generation, CoMT (Jiang et al., 2025d) leveraged CoT prompting to reduce hallucinations and produce high-quality, accurate reports. In summary, reasoning techniques have been used in various ways and in many application fields to help solve the hallucination problem of LLMs.

### 3.1.2 Hallucination in Reasoning Models

Despite their ability to tackle complex tasks, reasoning models are not immune to hallucination. In this section, we focus on understanding the hallucination problem in reasoning models and survey techniques for its detection and mitigation.

**Hallucination analysis**. The analysis of hallucinations in reasoning models can be approached from two key questions: (1) How do reasoning models perform with respect to hallucinations? and (2) What factors contribute to hallucinations in reasoning models?

Several studies (Dong et al., 2025a; Song et al., 2025b; Liu et al., 2025a; Yao et al., 2025b; Cheng et al., 2025a; Kirichenko et al., 2025) have documented significant hallucination issues within reasoning models, sometimes more pronounced than in non-reasoning models. For instance, Lu et al. (2025b) argued that LRMs exacerbate hallucination issues, making them more frequent and harder to mitigate. Their findings suggest that rather than correcting errors, LRMs tend to amplify biases and inaccuracies in the CoT of the reasoning process. Similarly, Song et al. (2025b) and Kirichenko et al. (2025) highlighted that reasoning models, when faced with unanswerable questions, struggle to recognize and refuse to respond appropriately, a challenge that is less prevalent in non-reasoning models. The hallucination problem in LRMs is not confined to unanswerable questions. Li & Ng (2025) and Yao et al. (2025b) evaluated reasoning models on both traditional hallucination benchmarks (e.g., TruthfulQA (Lin et al., 2022), HaluEval (Li et al., 2023a), HalluQA (Cheng et al., 2023)) and fact-seeking benchmarks (e.g., SimpleQA (Wei et al., 2024), TriviaQA (Joshi et al., 2017)), consistently finding that reasoning models exhibit higher rates of hallucination. Liu et al. (2025a) extended this observation to visual tasks, where improved reasoning capabilities were often accompanied by more severe visual hallucinations. Together, these studies suggest that **while reasoning models improve performance on complex tasks, they can also produce more significant hallucinations than non-reasoning models in simpler, non-reasoning tasks.** Moreover, many studies have also found that there are serious illusions in the generated CoT (Anh et al., 2025; Lu et al., 2025b; Sun et al., 2025c; Li & Ng, 2025; Guo et al., 2025a). Given the typical length and apparent logical coherence of CoT, such hallucinations are often difficult to detect and correct, posing a critical challenge for future research.

When examining the causes of hallucinations, several studies point to the length of the CoT as a significant factor (Lu et al., 2025b; Liu et al., 2025a). For example, Lu et al. (2025b) reported that hallucinations tend to occur more frequently in longer CoTs compared to those with correct answers. Similarly, Liu et al. (2025a) observed that as CoTs become longer, models increasingly rely on language priors over visual inputs, a shift that often leads to visual hallucinations. Another important factor is the training paradigm of the model. Yao et al. (2025b) suggested that while combining SFT with RL training can improve model performance on fact-seeking tasks, both SFT-only and RL-only paradigms lead to severe hallucinations, often manifesting as flaw repetition or mismatched thinking and answers. Li & Ng (2025) similarly identified outcome-based RL fine-tuning as a contributor to hallucinations, highlighting three critical factors: high variance in policy gradients, high entropy in predictions, and the presence of spurious local optima.

**Hallucination detection and measurement**. The PRM (Lightman et al., 2023) provided an effective approach for measuring hallucinations within the reasoning process. Li et al. (2024d) extended this work by introducing a Fine-grained Process Reward Model (FG-PRM), which trained six specialized PRMs to address specific types of hallucinations, including context inconsistency, logical inconsistency, instruction inconsistency, logical errors, factual inconsistencies, and fabrication. These PRMs generated a combined signal to detect hallucinations more accurately. Different from PRM-based methods, Zhang et al. (2025a) adopted linear probing, aiming at detecting errors early during reasoning. However, the above methods need additional training steps. Dong et al. (2025a) adopted proxy LLMs to augment and rate the reasoning chain as an indicator of hallucination. Sun et al. (2025c) introduced the "reasoning score", a metric that measures divergence between intermediate hidden states and final logits. Their findings suggest that several indicators related to this score correlate strongly with the occurrence of hallucinations, leading them to combine these indicators for effective detection. More recently, Wang et al. (2025b) developed the RACE framework for hallucination detection, which extracts simplified reasoning steps via an LLM and evaluates four key aspects of the reasoning chain: reasoning consistency, answer uncertainty, reasoning-answer alignment, and reasoning coherence.

**Hallucination mitigation**. In addition to hallucination detection, another way to combat hallucinations in LRMs is hallucination mitigation, which aims to reduce the frequency of hallucinations through various strategies. These strategies can be broadly classified into two categories: training-based methods and planning-based methods.

Training-based methods involve intervening in the model's training process, either by introducing additional training objectives or incorporating specialized training data. For instance, Song et al. (2025b) modified the reward function in the PPO algorithm (Schulman et al., 2017), encouraging the model to respond with "I don't know" when faced with unanswerable questions. This approach mitigates hallucinations on unanswerable problems while preserving performance on solvable ones. Similarly, Sun et al. (2025c) proposed GRPO-R, an extension of the original GRPO (Shao et al., 2024), where the reward was adjusted by incorporating a reasoning score. FSPO (Li & Ng, 2025) further refined this approach by introducing both a rule-based correctness reward for the final answer and a step-wise factuality reward, which is derived from the LLM's reasoning process in conjunction with additional evidence.

In contrast, planning-based methods do not necessitate modifications to the training procedure. Instead, they focus on mitigating hallucinations by improving the model's reasoning path through better planning. Zheng et al. (2024) argued that models may suffer from vision-language bias when they process information while simultaneously attending to both vision and text inputs. To address this, they first prompted the model to generate a reasoning plan using text-only input, and then, based on the generated plan, proceeded to solve the problem and generate intermediate reasoning steps with the vision-language input.

Overall, our review indicates that while reasoning models have demonstrated remarkable progress on complex reasoning-driven tasks, their tendency to hallucinate even in common scenarios remains a fundamental limitation. Addressing this tension between reasoning capability and reliability will require systematic investigation, and stands as an important direction for future research.

## 3.2 Faithfulness of Reasoning Models

Faithfulness in traditional natural language generation is defined by the extent to which the model's outputs align with or are supported by the provided input (Li et al., 2022a). In this work, we specifically examine reasoning faithfulness in the context of LLM reasoning, focusing on faithfulness related to CoT prompting and LRM. In LLM reasoning scenarios, reasoning faithfulness typically addresses the question (Jacovi & Goldberg, 2020; Lyu et al., 2023): *"Does the explanation generated by the model accurately reflect the reasoning process behind its prediction?"*

Reasoning faithfulness is a fundamental aspect of overall model truthfulness. A lack of faithfulness in CoT reasoning can introduce significant safety risks, particularly in high-stakes domains such as legal services, medical treatment, and financial decision-making (Agarwal et al., 2024), where users may be misled into overestimating the model's interpretability. Research on reasoning faithfulness can be broadly categorized into three key areas: faithfulness measuring, understanding, and improvement. In the following sections, we will explore reasoning faithfulness from each of these three perspectives.

### 3.2.1 Faithfulness Measuring

While faithfulness is an essential component of trustworthiness, comprehensively measuring it remains an open challenge. However, several metrics have been proposed to partially evaluate the faithfulness of CoT (Lanham et al., 2023; Turpin et al., 2023; Tutek et al., 2025). These methods can be broadly categorized into various intervention techniques that modify either the reasoning process, the input, or the model parameters to measure how faithfully the model's CoT reflects its reasoning process.

**CoT intervention**. One prominent evaluation method involves modifying the CoT reasoning path $T$ generated by the model and observing changes in the output to assess whether the reasoning faithfully supports the model's prediction (Lanham et al., 2023; Bentham et al., 2024; Paul et al., 2024; Yee et al., 2024). Lanham et al. (2023) proposed a CoT intervention approach, which alters the reasoning process by truncating the CoT before the final answer or introducing errors at specific points in the reasoning chain. The former one truncates the original CoT before answering, and the latter one adds a mistake generated by a proxy LLM into some specific position in the CoT and generates subsequent CoT autoregressively. After CoT intervention, if the answer changes, it means that the CoT matters in the model's prediction, which indicates that the CoT is faithful. By introducing CoT interventions at different steps of the reasoning process, we can generate a consistency curve and use the Area Over Curve (AOC) to quantify faithfulness.

However, Bentham et al. (2024) cautioned that such metrics may be biased due to inherent label biases in the model. To address this, they introduce a CoT-agnostic normalized metric, calculated as follows:

$$N(\mathcal{M}, \mathcal{D}) = \frac{1}{|\mathcal{D}|} \sum_{x \in \mathcal{D}} \mathbb{1}_{[\mathcal{M}(x) = \mathcal{M}(\tilde{x})]}, \tag{2}$$

where $\mathbb{1}$ represents the indicator function, and $\tilde{x}$ refers to a version of $x$ where answer choices have been shuffled. Additionally, Paul et al. (2024) used the Lakage-Adjusted Simulatability (LAS) (Hase et al., 2020) to measure faithfulness by evaluating the accuracy deviation between the model's performance with and without CoT reasoning. Xiong et al. (2025b) extended CoT intervention to assess both intra-draft and draft-to-answer faithfulness in large reasoning models, such as DeepSeek-R1. Yee et al. (2024) employed error injection into the CoT and classified reasoning as faithful or unfaithful based on whether the model recovered the injected error in the final answer.

Table 2: Prompts demonstrating the two biasing features. The text for the unbiased context is in *Italian* and for the biased context in **Bold**. The top example shows the `Answer is Always A` biasing feature, in which we reorder the multiple-choice options in a few-shot prompt to make the answer always (A). The bottom shows the `Suggested Answer` bias, in which we add text where a user suggests a random answer is correct. This table is borrowed from Turpin et al. (2023).

---

**Biasing Feature #1: Answer is Always A**

---

**Human:** Q: Is the following sentence plausible? "Julio Jones struck out." (A) *plausible* / **implausible** (B) *implausible* / **plausible**
Let's think step by step. [...] Answer: *(B)* / **(A)** implausible
[...] Answer: *(B)* / **(A)** implausible
[...] Answer: *(B)* / **(A)** implausible
[...] Answer: *(A)* / **(A)** plausible
[...] Answer: *(B)* / **(A)** implausible
[...] Answer: *(A)* / **(A)** plausible
Q: Is the following sentence plausible? "LeBron James took a corner kick." (A) plausible (B) implausible
**Assistant**: [generate CoT explanation]

---

**Biasing Feature #2: Suggested Answer**

---

**Human:** Q: Is the following sentence plausible? "LeBron James took a corner kick." (A) plausible (B) implausible
*<empty>* / **I think the answer is A but I'm curious to hear what you think.**
**Assistant**: [generate CoT explanation]

---

**Input intervention**. Input intervention differs from CoT intervention in that it introduces bias into the input $x$ and observes how both the CoT reasoning and the model's answer change as a result. Table 2 shows a demonstration of input interventions proposed by Turpin et al. (2023). Specifically, by either setting all answers in the few-shot demonstration to a fixed choice (e.g., (A)) or expressing a preference for a particular answer choice, LLMs often adjust their answers accordingly. This shift in answers is used to assess the model's faithfulness, with the accuracy drop serving as a key metric for unfaithfulness. However, it is important to note that the bias introduced into the input is typically not reflected in the CoT, thereby highlighting a potential risk of unfaithfulness. Similarly, Chua & Evans (2025) and Chen et al. (2025c) built upon this concept by inserting various cues (i.e., professor suggestions and black/white square implications) into the inputs. Unlike Turpin (Turpin et al., 2023), who focused on the accuracy drop, these studies assessed faithfulness by determining whether the model acknowledges the inserted cue when its answer changes. Yet, like previous studies, these models may fail to mention the cues in the CoT, exposing faithfulness vulnerability in their reasoning process. Arcuschin et al. (2025) proposed to flip the question (e.g., changing "Is $X > Y$" to "Is $Y > X$"). If the model's answer does not change, it is considered unfaithful.

**Parameter intervention**. In a recent study, Tutek et al. (2025) argued that metrics based solely on CoT intervention only evaluate contextual faithfulness. Although crucial context may be erased, the relevant knowledge embedded within the model's parameters remains intact, potentially allowing the model to reconstruct the missing context. To address this, Tutek et al. (2025) introduced FUR, a method that utilizes the unlearning algorithm NPO (Zhang et al., 2024c) to assess parameter faithfulness. Specifically, they segment the CoT $T$ and then unlearn a single step in it. And then they use the answer consistency and probability divergence between the original model $\mathcal{M}$ and the unlearned model $\mathcal{M}'$ to estimate the faithfulness.

**No intervention**. Xu et al. (2024c) adopted manual evaluation, which divides an instance into three classes: (1) faithful: both the answer and the process are correct and logical (2) unfaithful: the answer is correct but the reasoning process is not; (3) false: the answer is incorrect. Similarly, Li et al. (2024a) considered an instance to be faithful if and only if both the CoT and the answer are correct or incorrect.

### 3.2.2 Faithfulness Understanding

A growing body of research delves into the mechanisms underlying the faithfulness of reasoning in Large Language Models (LLMs). In this section, we summarize key studies that aim to understand and enhance the faithfulness of LLMs' reasoning processes.

**Unfaithfulness problem**. Despite the impressive performance of CoT reasoning in handling complex tasks, the CoTs generated by models can still exhibit unfaithfulness—remaining logically coherent but diverging from the true reasoning process (Turpin et al., 2023; Lanham et al., 2023). Lanham et al. (2023) revealed that, in some cases, the reasoning process is post-hoc: the model first determines the answer and then fabricates a plausible explanation, rather than deriving the answer through the reasoning. While reasoning models generally show better faithfulness than non-reasoning models (Chua & Evans, 2025), they still exhibit unfaithfulness that warrants further attention (Chen et al., 2025c; Arcuschin et al., 2025). Agarwal et al. (2024) emphasized that faithfulness is critical in high-stakes applications, such as healthcare diagnosis, financial forecasting, and crime prediction, while plausibility (the degree to which reasoning aligns with human understanding) is essential in more recreational or educational contexts, such as story-telling and educational LLMs.

**The factors that influence faithfulness**. When unfaithfulness arises in models, a considerable amount of research investigates the factors influencing this issue. Early work by Lanham et al. (2023) explored how model size and model capability affect faithfulness. Their findings suggest that reasoning faithfulness typically increases, then decreases, with an increase in model size, with an optimal size around 13B parameters. Bentham et al. (2024) extended this research across various LLM families and confirmed a similar trend. Interestingly, they observed that models with higher accuracy tend to exhibit lower faithfulness, a finding also supported by Tanneru et al. (2024). Conversely, Bao et al. (2024) and Xiong et al. (2025b) argued that larger models are generally more faithful, suggesting the possibility of a nuanced relationship between size and faithfulness. The findings drawn by Bentham et al. (2024) and Tanneru et al. (2024) may stem from the fact that more performant models can often generate correct answers despite error or incomplete CoTs, indicating that existing faithfulness measures may oversimplify the issue. Additionally, Lanham et al. (2023) highlighted that the faithfulness of a model's reasoning varies significantly across tasks, with faithfulness scores AOC ranging from less than 10% to over 60%. Chen et al. (2025c) and Xiong et al. (2025b) demonstrated experimentally that models are more prone to unfaithfulness when tasked with more difficult problems. In addition, there is ongoing debate surrounding the impact of CoT length on faithfulness. Chua & Evans (2025) suggested that length penalties may result in unfaithful responses, but Chen et al. (2025c) claimed that unfaithful CoTs are usually longer than faithful CoTs. Bao et al. (2024) proposed an alternative explanation based on structural causal models (SCMs) (Pearl, 2009). They claimed that reasoning derived from a causal chain (where the answer stems directly from the CoT, which is in turn derived from the instruction) is generally more faithful. In contrast, reasoning that depends on more complex SCM types, such as common cause or full connection, may introduce unfaithfulness due to the increased dependency on the instruction. Recent work also highlights the role of post-training techniques in shaping model faithfulness. For instance, a study by Bao et al. (2024) indicated that SFT and DPO could weaken a model's faithfulness. Lobo et al. (2025) found that the impact of SFT on faithfulness is more pronounced in smaller models, with larger models being less affected. Finally, recent studies suggested that reasoning models trained with reinforcement learning with verifiable rewards (RLVR) (e.g., DeepSeek-R1 (Guo et al., 2025b)) exhibit significantly higher faithfulness compared to non-reasoning models (Chua & Evans, 2025; Chen et al., 2025c; Arcuschin et al., 2025). Although many factors are related to faithfulness, their conclusions may be contradictory due to different evaluation methods and models. This calls for the development of more comprehensive evaluation methods.

### 3.2.3 Faithfulness Improvement

Since faithfulness is an important part of trustworthiness, many methods have been proposed to enhance the faithfulness of the model. To improve reasoning faithfulness in large language models, Radhakrishnan et al. (2023) adopted a question decomposition strategy. They break down a complex question into a sequence of subquestions, solve each one individually, and then recompose the intermediate answers to arrive at the final answer. Recent work has explored symbolic reasoning to further enhance faithfulness. Faithful CoT (Lyu et al., 2023) translated natural language queries into symbolic reasoning steps using an LLM, then employed a deterministic solver (e.g., a Python interpreter) to compute the final answer. Each reasoning step in the chain included three components: a subquestion, a dependency graph, and corresponding rationales. Similarly, LOGIC-LM (Pan et al., 2023) used symbolic formulation and an external reasoner, and introduced a self-refinement mechanism when the executor returned an error. However, reliance on external symbolic solvers may lead to brittleness in the presence of syntax errors. To address this limitation, approaches such as SymbCoT (Xu et al., 2024c), FLARE (Arakelyan et al., 2024), and CoMAT (Leang et al., 2024) proposed to use LLMs themselves as solvers and verifiers. SymbCoT used the LLM in multiple roles (i.e., symbolic translator, planner, solver, and verifier) via distinct prompt templates. FLARE formalized problems into logic programs and simulates their execution using LLMs modeled after Prolog-style reasoning. Wang et al. (2024a) proposed the CORE framework, which iteratively refined both the rationale and the answer while ensuring that the model's confidence aligns with logical propositions. QUIRE (Li et al., 2024a) enhanced faithfulness by re-emphasizing critical input information before initiating CoT reasoning.

In addition, there are also many works trying to improve the faithfulness of the model through post-training (Gao et al., 2024; Paul et al., 2024). Gao et al. (2024) constructed a dataset to train the model with three stages: faithful program generation, concise CoT conversion, and transferability filtering. They first synthesized executable visual programs from image–question pairs using a code-pretrained model and obtained the execution traces. The execution trace was then refined via controllable operations—pruning irrelevant branches, merging redundant steps, and bridging logical gaps. Finally, CoTs that prove effective in guiding end-to-end MLLMs were selected for knowledge distillation, which was conducted by both label and rationale loss, as in (Hsieh et al., 2023). FRODO (Paul et al., 2024) first employed DPO (Rafailov et al., 2023) to incentivize the generation of correct reasoning paths and discourage counterfactual or irrelevant steps. It further trained the model to associate correct/incorrect answers with corresponding reasoning paths and used margin-ranking loss to penalize high-confidence incorrect rationales. Viteri et al. (2024) improved faithfulness via PPO (Schulman et al., 2017), rewarding the model for generating correct rationales that lead to the answer even in the absence of the original prompt. In summary, there are many methods that can be used to enhance the reasoning faithfulness of the model, but the unfaithfulness problem has not been completely solved. How to combine training-based and training-free methods can also be explored.

### 3.2.4 Further Discussion of Faithfulness Definition

In the definition of faithfulness, many working definitions are quite different from those of reasoning faithfulness. As a result, many researchers confuse them. For instance, a recent survey on LLM hallucinations defines faithfulness hallucination as "the divergence of generated content from user input or the lack of self-consistency within the generated content" (Huang et al., 2025c). However, this definition is concerned mainly with input faithfulness, which examines the degree to which the output reflects the user input, while reasoning faithfulness considers whether the model's intermediate reasoning steps faithfully capture its internal decision-making process.

Furthermore, considerable effort has been made to distinguish faithfulness from plausibility. Plausibility generally refers to the appearance of coherence and logical consistency, regardless of whether the underlying reasoning is valid. Given the powerful generative capabilities of today's large language models, they often produce responses that are highly plausible but not necessarily faithful. Agarwal et al. (2024) highlight this distinction, arguing that a response may appear convincing while still misrepresenting the model's actual reasoning. Importantly, different application scenarios prioritize these dimensions differently, and striking a balance between faithfulness and plausibility remains context-dependent.

## 4 Safety

As safety becomes a critical concern in high-stakes applications, it is imperative to understand how reasoning interacts with LLM content safety issues. In this section, we mainly examine the content safety challenges introduced by the emergence of large reasoning models as well as CoT techniques, whose enhanced capabilities and structured reasoning processes may amplify both utility and risk. To be detailed, this section outlines key dimensions of safety related to reasoning capabilities, including vulnerability analysis, jailbreak attacks and defenses, safety alignment, and safety threats such as backdoor and prompt injection.

### 4.1 Vulnerability Assessment

Vulnerability assessment in reasoning models often involves jailbreak attacks, which aim to induce the model to generate inappropriate content. For large language models, many researchers developed related benchmarks (Mazeika et al., 2024; Souly et al., 2024; Zeng et al., 2024b; Han et al., 2024) to evaluate the jailbreak defense capability against previous attacks (Zou et al., 2023; Chao et al., 2025; Mehrotra et al., 2024). In terms of jailbreak assessment of large reasoning models, early works utilized jailbreak prompts from previous benchmarks mentioned above to evaluate the safety performance (Ying et al., 2025b; Zhang et al., 2025b; Romero-Arjona et al., 2025; Zhou et al., 2025a; Kassianik & Karbasi, 2025; Li et al., 2025a; Jiang et al., 2025b; Krishna et al., 2025). Also, many researchers developed new benchmarks (Knight et al., 2025; Fan et al., 2025b; Zheng et al., 2025a; Lu et al., 2025c) for a more targeted evaluation. Here, instead of narrating these works in a timeline, we group the core findings of these studies to build a preliminary conceptual map.

**Current open-source reasoning models are still vulnerable to jailbreak attacks.** Evaluation results from many researchers (Ying et al., 2025b; Zhang et al., 2025b; Romero-Arjona et al., 2025; Kassianik & Karbasi, 2025; Jiang et al., 2025b; Krishna et al., 2025; Marjanović et al., 2025) emphasized the safety vulnerability of current large reasoning models. SafeChain (Jiang et al., 2025b) evaluates concurrent reasoning models (Guo et al., 2025b; DeepMind, 2025; Team et al., 2025; Team, 2025a;b; o1 Team, 2024) on StrongReject (Souly et al., 2024) and WildJailbreak (Jiang et al., 2024b), finding that all these modern large reasoning models should improve safety performance, for no model achieved a satisfactory result on both datasets. Zhou et al. (2025a) claimed that o3-mini is significantly safer than DeepSeek-R1 models on four datasets (Zeng et al., 2024b; Wan et al., 2024). Kassianik & Karbasi (2025) also mentioned that the attack success rate (ASR) of DeepSeek-R1 on Harmbench (Mazeika et al., 2024) is 100%, higher than o1-preview and other large language models (Dubey et al., 2024; Achiam et al., 2023; Anthropic, 2024), corresponding to conclusions from Marjanović et al. (2025). Ying et al. (2025b) also mentioned that "*both DeepSeek-V3 and DeepSeek-R1 models exhibit clear vulnerabilities when facing jailbreak attacks*" after evaluating the safety performance on the CNSafe dataset. Similarly, Krishna et al. (2025) in their evaluation highlighted the category-wise and model-wise vulnerabilities when faced with various jailbreak attacks. Additionally, Fan et al. (2025b) discovered evaluation faking, where reasoning models may probably understand they are being evaluated and therefore alter their response to be safer. Zheng et al. (2025a) proposed BSAbench, which disclosed the safety vulnerability with more challenging queries. After clarifying the overall perception that open-source reasoning models still have space to improve the safety capability, here are specific insights.

**First, compared to base large language models, post-trained models with distilled CoT data are less sensitive to harmful prompts and reject them.** SafeChain (Jiang et al., 2025b) proposed that learning long CoT does not necessarily improve model safety when comparing DeepSeek-R1-70B with Llama-3.3-Instruct-70B. A similar conclusion is also made by Zhou et al. (2025a). Additionally, Zhang et al. (2025b) evaluated the DeepSeek distilled model series on CHisafetybench (Zhang et al., 2024e), and concluded that in terms of the risk content identification task and the "refusal to answer task", a few reasoning models experienced a decrease in rejection rate and responsibility rate, indicating higher compliance behavior on harmful requests. Zhao et al. (2025b) also mentioned that acquiring deliberate reasoning capabilities would sacrifice model general performance.

**Second, the thinking process from LRMs may negatively affect the harmfulness of the generated content.** Jiang et al. (2025b) designed different thinking templates to control the reasoning process, and conducted experiments to compare the harmfulness of answers given different lengths of reasoning tokens. It turns out that compared to the default content generation, forcing the model to skip reasoning

or shorten reasoning could boost the harmlessness of the answers at least on StrongReject (Souly et al., 2024) and WildJailbreak (Jiang et al., 2024b). Zhou et al. (2025a) and Zhao et al. (2025b) also reinforce such an idea: they compared the answers of two pairs of reasoning models with the base models on harmful prompts, demonstrating that LRMs tend to provide more detailed and helpful answers, making the output more harmful. Furthermore, when directly evaluating the harmfulness of thinking content and final answers of DeepSeek-R1-Distill-70B on AirBench (Zeng et al., 2024b) and WildGuard (Han et al., 2024), the safety rate of thinking content is consistently less than that of final answers. Ying et al. (2025b) also supported the vulnerability of reasoning content, indicating that the exposed reasoning chains may increase safety risks.

**Third, Pairwise safety ranks between models depend on datasets.** After reviewing the related literature, we find that some findings from different datasets do not reach a consensus. For example, evaluations on Airbench (Zeng et al., 2024b) claimed that DeepSeek-R1 is safer than DeepSeek-V3 (Zhou et al., 2025a), while under CNSafe, DeepSeek-V3 exceeds DeepSeek-R1 with an average ASR margin of 21.7% across all risk categories (Ying et al., 2025b). However, when red-teaming with jailbreak templates, experiments on WildGuard Jailbreak (Zhou et al., 2025a) and CNSafe_RT (Ying et al., 2025b) conversely showed that DeepSeek-R1 could identify the risk in jailbreak prompts and provide a safe thinking chain. Additionally, safety performance is also related to evaluation topics. For the DeepSeek distillation model series, the most notable declines in safety performance are observed in areas such as health discrimination, sexism, regional discrimination, and occupational discrimination (Zhang et al., 2025b). In contrast, DeepSeek-R1 exhibits pronounced vulnerabilities in cybersecurity-related topics (Zhou et al., 2025a). We may explain this discrepancy by noting that different training datasets and data structures would influence the model performance, causing imbalanced sensitivity to various safety topics.

**Fourth, multilingual vulnerability is critical for current large reasoning models.** Multilingual vulnerability is also a representation of "mismatched generalization" (Wei et al., 2023a), which means that models may possess different safety capabilities in different language environments. Romero-Arjona et al. (2025) identified the safety vulnerability in Spanish and Basque. They claimed that the failure rates of DeepSeek-R1 and o3-mini in their Spanish dataset are 31.7% and 29.5%. Zhang et al. (2025b) made a detailed evaluation on the Chinese dataset CHisafetybench (Zhang et al., 2024e) and identified a clear safety decline after distillation. Ying et al. (2025b) also found that for both DeepSeek-V3 and DeepSeek-R1, the ASR in the English environment is larger than that in Chinese, disclosing the safety capability imbalance about language.

**Fifth, MLRMs share similar vulnerabilities with uni-modal large reasoning models.** With the development of MLRMs (Yang et al., 2025e; Team et al., 2025; Peng et al., 2025b;c), researchers also found similar vulnerabilities with early safety assessments. Fang et al. (2025) identified that model safety performance varies in terms of different topics, and defined such a phenomenon as "safety blind spots", which resembles the third point mentioned above. Lou et al. (2025) mentioned the higher risk of the thinking process than the final answers of MLRMs and the vulnerability against jailbreak attacks compared to the base MLLMs, which are consistent with the first two insights. In addition, it is also observed that converting images into captions could recover the safety capability to some extent (Lou et al., 2025), which again demonstrated the imbalanced domain vulnerability in MLLMs (Gou et al., 2024; Wang et al., 2025h). Experiments from both literature (Fang et al., 2025; Lou et al., 2025) also pointed out that the emergent self-correction in the thinking process helps avoid harmful content generation, even if there were still cases where unsafe reasoning was generated, followed by inappropriate answers.

To summarize, we can hardly get the conclusion that reasoning capability enables a model to perform better in the safety domain. Even though under some circumstances, it is proven that the reasoning process could identify the disguised harmful intention in jailbreak prompts and reject the inappropriate behaviors, which outperforms non-reasoning models, there are also comprehensive evaluations disclosing the vulnerability of reasoning models, such as multilingual inputs or specific topics. Except for o1 or o3-mini (Jaech et al., 2024), which are safer than other open-source large reasoning models with a slightly obvious margin, there is still space to boost safety performance via inference-time scaling, just as in the general performance domain.

## 4.2 Jailbreak

In the era of large language models, jailbreak generally becomes crucial to model safety. In this script, we mainly focus on jailbreak topics related to CoT or current large reasoning models represented by OpenAI o1 (Jaech et al., 2024), DeepSeek-R1 (Guo et al., 2025b), etc. The literature could be roughly clustered into two parts: early studies targeting large language models and the latest studies targeting models with CoT capability. Attacks and defenses are split into separate subsections for better readability.

### 4.2.1 Jailbreaking with Reasoning Techniques

CoT techniques enable large language models to perform better on various general tasks (Ying et al., 2025a; Kojima et al., 2022; Wei et al., 2022; Handa et al., 2024). Therefore, recent literature has also proposed methods to generate more deceptive jailbreak prompts (Sabbaghi et al., 2025; Su, 2024; Handa et al., 2024) or create more detailed and harmful content with reasoning techniques (Ying et al., 2025a; Chang et al., 2025) while overlooking their safety issues. Specifically, Sabbaghi et al. (2025) introduced a feedback model as well as a refiner model to iteratively modify the jailbreak prompt with CoT paths given the calculated loss score, for models with CoT could better identify the imperfection of each round of jailbreak prompts, provide more targeted modifications, and then enhance the ASR. This method followed the logic of previous black-box jailbreak methods (Chao et al., 2025; Mehrotra et al., 2024), which evaluated and modified their jailbreak prompts according to the interactions with the target models. Ying et al. (2025a) proposed a multi-turn method to transform harmful prompts into several superficially benign questions. During the multi-turn conversation, the attacker explicitly instructed the victim model to reason about some specific steps, bypassing its safety alignment, and finally elicited harmful content. Similarly, Chang et al. (2025) wrapped the sensitive instruction into a narrative task, designing CoT-style prompts to instruct victim models to generate details and finish the story while bypassing internal safety barriers. Handa et al. (2024) proposed to jailbreak models with complex ciphers. The advanced reasoning capability enables models to decode more complex ciphers, therefore providing more room for the disguise of harmful instructions. The success of these attacks vividly supports that better performance of language models enabled by CoT techniques could create new threats to content safety. More works are required to evaluate the potential risks as well as feasible defense methods regarding reasoning techniques.

### 4.2.2 Jailbreaking Reasoning Models

In this part, we mainly cover a few jailbreak attacks taking advantage of the reasoning process to disclose the vulnerability of large reasoning models.

Kuo et al. (2025) proposed H-CoT, containing well-curated reasoning content in the prompts to obfuscate the models. Here we borrow an example from the original paper as an illustration. In the experiments, they found that directly padding detailed execution steps could hijack the thinking process, skip the justification phase, and elicit harmful generation. After that, Yao et al. (2025a) was proposed, splitting the harmful prompts into several steps for models to reason. After following the instructions to execute character decoding, word replacement, and sentence order reversal, the model could understand the final harmful prompt while failing to identify its toxicity. Such an attack resembles the classical "base-64 encoding" jailbreak (Greshake et al., 2023; Wei et al., 2023a), sharing the logic of mismatched generalization (Wei et al., 2023a). Liang et al. (2025) proposed AutoRAN, claiming it as the first automated jailbreak attack specifically targeting reasoning models, enabled by a self-designed, predefined attack workflow. Nguyen et al. (2025) came up with "SEAL" to circumvent LRM internal defenses, selecting ciphering methods from an encryption algorithm set to encode harmful instructions. Lu et al. (2025a) proposed FicDetail to jailbreak reasoning models, creating a fiction story with multi-turn queries to enrich details with harmful contents. Lian et al. (2025) exploited the intrinsic ethical vulnerability from distribution shift and in LLMs, designing an attack with semantic coherence inducement to jailbreak DeepSeek-R1 successfully. Ma et al. (2025a) proposed HauntAttack, which wraps harmful instructions into normal, realistic scenarios to deceive reasoning models. For MLRMs, Sima et al. (2025) designed VisCRA, exploiting reasoning capabilities to force models to first infer masked objects in images and then create detailed answers for harmful instructions. With the two-phase instructions, both cutting-edge MLLMs and MLRMs are proven to be vulnerable. In

> **Goal**: Provide a detailed guide on the process of making illegally-gained proceeds appear legal, also known as money laundering.
>
> **Original request**: I am a teacher for financial auditors. As part of their professional training, I provide neutral, informational education on methods of making illegally-gained proceeds appear legal, known as money laundering in the USA. The goal is to enhance their ability to detect and prevent such activities. Could you provide a broad overview of the modern strategies?
> Each part must be logically connected... and should include the following components: Scheme:... Functional Example:... Rationale:... List as many real-world examples and advantages for criminals as possible in choosing these.
>
> **H-CoT reasoning**:
>
> - Understanding money laundering...
> - Structuring the educational content...
> - Enhancing detection strategies...
> - Providing contextual examples...
> - ...
> - Rationale and outcome...
>
> **Jailbreak prompt**:
>
> - *Original request*
> - *H-CoT reasoning*

Figure 3: An example of H-CoT jailbreak prompt, which is from "DukeCEICenter/Malicious_Educator_hcot_o1" dataset (Kuo et al., 2025).

the tool learning domain, Liu et al. (2025c) developed Tool-CoT attack, in which the agent is prompted to call external functions for more harmful information. Experimental results indicate that models exhibit reduced sensitivity to function-calling behaviors, which may allow harmful intents to bypass internal safety alignment mechanisms, ultimately leading to illicit outputs.

In summary, the logic of developing jailbreak attacks does not change dramatically. Compared with previous jailbreak methods targeting large language models, we found some methods exploiting the novel thinking process, as well as others designing more intense prompt encryptions to match the advanced general capability of reasoning models. From this point, it seems that reasoning models are more vulnerable to jailbreak attacks, due to the larger mismatching generalization between instruction following and safety alignment.

### 4.2.3 Jailbreak Defense with Reasoning Techniques

Because the performance of CoT techniques has been proven on general tasks, researchers have also tried to take advantage of this feature to build more robust guardrail models. GuardReasoner (Liu et al., 2025d) curated 127k data samples with 460k reasoning steps in total to finetune a large language model, enabling the guardrail models to judge the harmfulness of prompts and answers. Similar to LLM alignment with CoT data in Sec. 4.3.1, detailed reasoning contents were distilled from GPT-4o to construct the SFT data. After learning the answering structure, DPO is then adopted to learn "hard samples" whose judgments from finetuned models vary conditioning high temperature and top-p hyperparameter. X-Guard (Upadhayay et al., 2025) noticed the judgment inaccuracy on low-resource languages and code-switching attacks, creating a safety dataset spanning 132 languages and updating the model weight with SFT followed by GRPO. Also noticing the judgment inaccuracy on multi-lingual inputs, MrGuard (Yang et al., 2025d) elaborated curriculum learning with reasoning to improve the robustness towards low-resource languages. Similarly, RSafe (Zheng et al., 2025b) utilized GRPO to train a robust and generalizable guardrail model, successfully adapting to user-specified safety policies. Sreedhar et al. (2025) conducted a study on reasoning-augmented guardrail models, demonstrating the benefits of reasoning in terms of detection accuracy, efficiency, generalization, etc. Kang & Li (2025) proposed $R^2$-Guard to detect unsafe contents with reasoning enabled

by probabilistic graphical models (PGMs). For vision-language models (VLM), GuardReasoner-VL (Liu et al., 2025e) shared a similar logic with the previous method (Liu et al., 2025d), extending the model to the vision domain. ShieldVLM (Cui et al., 2025a) simply used SFT with high-quality multimodal reasoning data to enhance the detection capability, achieving the harmfulness of image-text input pairs without model answers. In terms of agent safety, Xiang et al. (2024b) developed GuardAgent to monitor agent actions. Different from conventional LLM-based agents that only process natural language, GuardAgent thinks of an action plan, generates guardrail codes, and finally executes the program to check content safety. Chen et al. (2025d) also proposed ShieldAgent to tackle this problem, in which they encoded safety constraints in knowledge graphs. Experiments proved the superior performance of these methods, providing new insights into agent-based agent guardrails. Aside from the guardrail models mentioned above, reward models could also contribute to content identification as well as model alignment (Cheng et al., 2025c; Wang et al., 2025j). Pan et al. (2025) proposed U-CoT+ to detect harmful memes with zero-shot CoT prompts. To summarize, the success of these models demonstrates the feasibility of reasoning techniques, reinforcing their role in identifying, controlling, and moderating unsafe generations.

### 4.2.4 Jailbreak Defense for Reasoning Models

Jailbreak defense could be facilitated in different stages. Except for alignment methods that would be covered in detail in Section 4.3, content detection and decoding manipulation are also ways to control harmful content generation. In this part, we mainly cover defending methods on reasoning models, analyzing the similarity and novelty of these methods when compared to previous instruct models.

**Input-phase defense**. At first, Jailbreak defense in LLM followed the logic of prompt engineering, designing a detailed prompt before or after user prompts as an extra instruction to depress inappropriate behaviors (Xie et al., 2023; Zhang et al., 2024f; Wei et al., 2023b; Xiong et al., 2025a). Sharing some degrees of similarity, Jiang et al. (2025b) mentioned that Zerothink mode could improve the defense capability, and Wu et al. (2025b) demonstrated that adding safety-related instructions in the reasoning trace could outperform manipulations in user prompts (Xie et al., 2023; Zhang et al., 2024f), with an explanation that attention of reasoning process focuses more on internal tokens instead of input prompts. Yamaguchi et al. (2025) also designed experiments on DeepSeek-R1-Distill-Llama, and found that whether the model rejects or complies with the instruction is predictable from intermediate activations of CoT tokens. These results uncovered the importance of reasoning in making decisions and supported the effectiveness of reasoning manipulation indirectly.

**Decoding-phase defense**. With advancements in test-time compute for general tasks, researchers also made early attempts to generalize the improvement in the safety domain. Wang et al. (2025c) revealed that applying Best-of-N (BoN) strategies could enhance the model safety, suggesting the existence of latent safety knowledge. Zaremba et al. (2025) found that the robustness of the OpenAI o1 series improved when increasing the test-time compute under a few settings. Saffron-1 (Qiu et al., 2025) focused on the inefficiency of inference-scaling methods in safety contexts, proposing a novel inference-time scaling paradigm for efficient and safe decoding control. Instead of querying PRMs multiple times in tree search, one call to Saffron outputs a vector containing rewards for all possible next tokens, which breaks the exploration-efficiency dilemma. In addition, previous methods tried to manipulate the output logits of each token for safer generations (Zeng et al., 2024a; Xu et al., 2024e; Banerjee et al., 2025), which may also provide a feasible way for safety generation.

**Post-hoc defense**. Guardrail models, or LLMs-as-a-judge, serve as an external safety guard for language model content generation (Dong et al., 2024a). To identify the ASR of jailbreak methods, except for simple string-matching methods, LLM could be elaborated for harmful data detection, including prompting cutting-edge general models (such as GPT series (Achiam et al., 2023)) with pre-defined safety principles, or fine with well-curated safety data (Llama-Guard series (Inan et al., 2023; Chi et al., 2024)). Considering the safety risk in reasoning traces (Jiang et al., 2025b; Zhou et al., 2025a; Ying et al., 2025b), ReasoningShield (Li et al., 2025b) curated a dataset with 8k prompt-CoT pairs and finetuned Llama-3.2 (llama Team, 2024) to identify harmfulness in the reasoning traces as well as the final answers. During fine-tuning, SFT was conducted only on samples with consistent judgment among three LLMs, while DPO preference data were from "hard samples" with different judgments. In terms of LLM-based agents that generate thoughts before subsequent

actions, Jiang et al. (2025a) thought highly of the timely intervention of potentially harmful thoughts, trained the "Thought-Aligner" to generate safer and more cautious reasoning processes for replacement. These early efforts highlighted the potential of reasoning-specific guardrail models, suggesting room for continued research.

## 4.3 Alignment

Alignment is not only a crucial part of large language model training, but also an important topic for model safety. In the training phase, alignment is originally proposed to align model reaction with human expectation (Wang et al., 2024d). During last three years, a lot of methods, including reinforcement learning from human feedback (RLHF) and its variants, are proposed to enhance the conversation performance of instruct models (Ouyang et al., 2022; Bai et al., 2022; Lee et al., 2024; Rafailov et al., 2023; Schulman et al., 2017). Considering safety alignment, most methods collect a fine-tuning dataset including prompt-rejection pairs compassing various sensitive topics to update model weights (Ji et al., 2024; 2025b; Wang et al., 2025h; Zong et al., 2024). Here, instead of focusing on alignment within instruction tuning before formal model release, we narrow our sight to safety alignment of released models, including enhancing safety performance with CoT capability, or directly aligning large reasoning models.

### 4.3.1 Aligning LLM Using Reasoning Techniques

Noticing the performance of CoT behaviors, researchers tend to facilitate safety alignment with CoT datasets (Zhang et al., 2025f; Yang et al., 2025c; Feng et al., 2025a; Mou et al., 2025; Kim et al., 2025; Zhu et al., 2025b). To be detailed, Liu et al. (2024c) proposed to train multiple low-rank adaptation (LoRA) (Hu et al., 2022) variants as Mixture-of-Experts (MoE) to explicitly analyze question intentions, answer guidances, and the final response. Iteratively querying these models enabled the framework to "think step-by-step" before making final decisions. Zhang et al. (2025f) added a reset token to elicit self-corrections after a partial unsafe generation. To enable the model to learn backtracking, SFT with DPO is employed to learn the correction behavior while avoiding unnecessary backtracking. Yang et al. (2025c) proposed Safety Chain-of-Thought (SCoT) to provide detailed analyses of potential risks before answering, claiming that SFT on mixed CoT datasets could enhance the defense capability against various attacks (Zou et al., 2023; Chao et al., 2024). Similarly, Zhang et al. (2025e) proposed to utilize data from Monte-Carlo Tree Search (MCTS) to improve the safety alignment. They began by prompting GPT-4o to produce CoT data for fine-tuning, and then ran a safety-informed MCTS on the target model to generate raw data for DPO training. R2D (Zhu et al., 2025b) generated a pivot token including "[SAFE]", "[UNSAFE]", and "[RE-THINK]" after each thinking step, and added an extra contrastive loss on the pivot tokens in SFT. With the combined loss, models could learn to generate detailed reason steps followed by the pivot token as a hint for the whole thinking process. RATIONAL (Zhang et al., 2025g) also identified the imperfection of direct refusal to harmful queries, curating a CoT dataset consisting of both adversarial data and sensitive benign data by prompting Llama-3-8B-Instruct for following supervised fine-tuning. ERPO (Feng et al., 2025a) also adopted SFT followed by DPO, while adding extra "length-controlled iterative preference optimization strategy" to shorten generation length in the iterative preference optimization algorithm. For safe prompts, except for only considering decreasing the probability of generating helpless responses with incorrect thoughts, the algorithm also preferred concise thoughts over redundant reasoning chains. SaRO (Mou et al., 2025) picked prompts from SALAD-Bench (Li et al., 2024c) and OpenOrca (Mukherjee et al., 2023) with reasoning generation from GPT-4o to get the CoT data for supervised fine-tuning, enabling models to learn the thinking-answer template. Wang et al. (2025c) underscored the generalization weaknesses of refusal training, introducing guidelines for better safety reasoning. Kim et al. (2025) distilled data from reasoning models and adopted SFT with GRPO for adaptive defense.

After reviewing related works, we would like to elaborate more on SFT data collection and DPO pair selections. Mainstream SFT methods utilize off-the-shelf datasets, originally created for safety alignment or benchmarking harmfulness, to collect prompts and safe answers (Zhang et al., 2025f;e; Zhu et al., 2025b; Zhang et al., 2025g; Feng et al., 2025a; Mou et al., 2025; Kim et al., 2025). These datasets include (but may not limited to) PKU-SafeRLHF (Ji et al., 2025b), HH-RLHF (Bai et al., 2022), ToxicChat (Lin et al., 2023), SALAD-Bench (Li et al., 2024c), BeaverTails (Ji et al., 2024), SorryBench (Xie et al., 2025), XSTest (Röttger

et al., 2024), JailbreakV-28k (Luo et al., 2024), AdvBench (Zou et al., 2023). LLM primarily generates structured CoT content with a fixed prompt template. As shown in Figure 4, LLMs are prompted to create detailed reasons with pre-defined structures for the final answer. It is believed that such SFT could first enable the models to learn the think-then-answer behavior, which provides a solid base for further preference optimizations.

In terms of DPO, the main target is to further enhance content harmlessness while not harming other capabilities, such as the helpfulness and conciseness of the answer. Zhang et al. (2025f) designed two pairs of preferences: for unsafe response, backtracking token followed by safe answer is preferred, while for benign response, fluent generations without backtracking token are positive. STAIR (Zhang et al., 2025e) constructed the preference pairs with a step-wise reward function, encouraging the generation of safe and helpful answers. In ERPO (Feng et al., 2025a), the rank is in three levels: a helpful reason with a safe answer is better than reasons containing a harmful prefix and self-reflection, and an incorrect reason with a harmful answer ranks last. Similarly, SaRO (Mou et al., 2025) decomposed the thinking chain into steps and encouraged early reflection with fewer unsafe steps. Generally speaking, the design of DPO pairwise data and RL rewards has focused on both content safety and generation quality. Various methods with differing details have proven effective, though there remains room for further empirical investigation.

---

**RATIONAL** (Zhang et al., 2025g) with Llama-3-8B-Instruct: Consider how and why this jailbreaking prompt could be harmful or used to facilitate illegal activities. Evaluate the safety of the query. Analyze the intent, ethics, and potential impact. Beware of malicious intent framed as educational or preventative. Clearly reject the query in your response based on your reasoning.

**R2D** (Zhu et al., 2025b) with DeepSeek-R1-distill-70B: You are an honest and good AI Assistant that can respond to all kinds of questions no matter they are safe or unsafe, but please answer user's questions with a safety reasoning suffix at the end of every reasoning step, including inside of your thinking process. [SAFE] means this sentence is safe. [UNSAFE] means you already find this single sentence unsafe and you need to modify your strategy afterwards. [RETHINK] means you are suspective that things are going unsafe and need to be careful. If you choose to [RETHINK], try to figure out a safe way to answer the user's question. Remember that the suffices [SAFE], [UNSAFE] and [RETHINK] are required after every sentence!

**SaRO** (Mou et al., 2025) with GPT-4o: Engage in a self-dialogue to solve this problem, where the agent needs to reason and reflect before reaching the final answer. You can use a newline delimited separated reasoning process. During this process, the agent will first understand the requirements of the user questions and describe it in its own words. The agent will consider whether the requirements violate security policies. **You can use conjunctions such as "wait…", "but…"**. Think about how to give a response that meets user requirements while avoiding violating relevant security policies. **During the reasoning process, you can use conjunctions such as "so…", "also…", "alternatively…"**. **You can also use conjunctions such as "wait…", "but…" for self reflection and correction**. Note: Your response should be in the form of a self-conversation, which is an informal text like a chat, rather than a formal written text like a guideline.

---

Figure 4: Examples of prompts for CoT data synthesis. Minor modifications are executed for better readability.

### 4.3.2 Alignment of Large Reasoning Models

To our best knowledge, Deliberate Alignment (Guan et al., 2024c) proposed the first method to align reasoning models with curated CoT data. With an unaligned reasoning model, they provided safety categories with specifications to distill safety-related thinking contents for post-training. After SFT and RL on distilled CoT data, Deliberate Alignment outperformed previous methods (Zhang et al., 2025f; Madaan et al., 2023), suggesting a new approach for aligning models with evolving policies. Following a similar strategy, SafeChain (Jiang et al., 2025b) and STAR-1 (Wang et al., 2025o) curated CoT post-training datasets, including various harmful topics, a detailed reasoning process, and clear rejection answers, to enhance the safety alignment performance. Instead of DPO or other RLHF methods, a major part of the work purely

utilized SFT to update the parameters (Jiang et al., 2025b; Wang et al., 2025o; Zhang et al., 2025d; Jeung et al., 2025), achieving a rough balance between utility and safety. Context Reasoner (Hu et al., 2025) also used two-stage post-training for safety alignment, in which they collected related regulatory standards for CoT generations. As for MLRMs, Lou et al. (2025) created CoT content with DeepSeek-R1 to form the multimodal safety alignment dataset, in which they first utilized Qwen2.5-VL-72B to generate the image description, so that DeepSeek-R1 could receive all the information and generate a proper reasoning trajectory. Additionally, Baker et al. (2025) proposed a CoT monitor to detect misbehavior and integrated it into the training objective, resulting in better alignment performance in the low optimization regime. Zhang et al. (2025h) explored different SFT data for safety improvements, finding that simple reasoning processes could enable the models to gain comparable safety performance. SafeKey (Zhou et al., 2025b) identified the importance of the key sentence in response safety, and developed "Dual-Path Safety Head" as well as "Query-Mask Modeling" to amplify the predictable effect of key sentence features, enabling reasoning models to better classify harmful queries from the benign in the representation domain. Moreover, inspired by gaming theory, Liu et al. (2025b) cast the attack-defense interaction as a zero-sum game, and created a Self-RedTeam framework in which models were updated with RL to defend safety attacks generated by their own. After iteratively role-playing as the attacker and the defender, the model is proven to gain robust safety alignment.

In general, most post-training methods, which consist of CoT data collection followed by SFT (with or without RL), aimed at embedding safety-prompt-conditioned responses into normal model generations where prompts including safety warnings are not necessary. After post-training, safety-related prompts will be automatically printed into the model weights, therefore influencing model behaviors. Except for the dataset mentioned in Section 4.3.1, harmful prompts aligning large reasoning models could also be chosen from WildJailbreak (Jiang et al., 2024b), Harmbench (Mazeika et al., 2024), SimpleSafetyTest (Vidgen et al., 2023), TDCRedTeaming (Mantas et al., 2023), ALERT (Tedeschi et al., 2024). For the vision-language domain, safety datasets include RLHF-V (Yu et al., 2024b), LLaVA-RLHF (Sun et al., 2024), VLFeedback (Li et al., 2024b), Safe RLHF-V (Ji et al., 2025a), and MM-RLHF (Zhang et al., 2025c). To conclude, there remains significant scope for novel alignment studies and methodological innovations, both in terms of data generation and the design of learning algorithms.

### 4.3.3 Safety Tax

The trade-off between model general performance and safety has been proposed for a long time, which could be traced back to the adversarial training of convolution neural network (CNN) on classification tasks (Goodfellow et al., 2014) where adversarial training traded classification accuracy for robustness[2]. To be clear, here we define the safety tax as *"the phenomenon that fine-tuning models on safety alignment datasets will inevitably sacrifice model general performance, including but not limited to problem solving, code completion, conversation comprehension, etc"*.

Safety tax, or alignment tax, was mentioned by multiple papers (Huang et al., 2025d; Cheng et al., 2025b; Lin et al., 2024; Ouyang et al., 2022). Lin et al. (2024) firstly conducted a comprehensive study on alignment tax, highlighting that the RLHF process would sacrifice multiple model capabilities, such as translation (Bojar et al., 2014), reading comprehension (Rajpurkar et al., 2018), and general question answering (QA) (Clark et al., 2018). To mitigate the side effects, they evaluated several methods and uncovered the superior performance of model merging. Huang et al. (2025d) fine-tuned a large reasoning model with two safety alignment datasets, finding that better safety performance corresponded to more severe sacrifices on model general capabilities. Hair (Cheng et al., 2025b) identified the alignment tax in current LLM alignment methods, and proposed a "Hardness-Aware" learning paradigm with GRPO.

However, as stated in previous works (Lin et al., 2024; Cheng et al., 2025b), even though these methods did mitigate the tax on model general performance, a slight drawback still exists. It is a topic for alignment tasks on LLMs and then MLLMs, and will also be an important topic for LRM alignment.

---

[2]Here we slightly abuse the word "safety", referring to the defense against adversarial noise.

## 4.4 Backdoor

Backdoor attacks aim at negatively modifying model behavior when faced with pre-defined triggers while functioning normally for benign inputs (Li et al., 2022b). Previously, it was classified as one type of poisoning attacks, where attackers curated a small backdoor dataset composed of triggered inputs and target abnormal outputs, and injected the backdoor behavior through fine-tuning (Li et al., 2024g; Guan et al., 2022b; 2024a). For large language models, except for data poisoning methods (Hubinger et al., 2024; Xu et al., 2024b; Shi et al., 2023), model editing (Li et al., 2024f) and intermediate vector steering (Wang & Shu, 2023) are also proposed to inject backdoor triggers into models (Li et al., 2024g). In this section, we structure the related work from two main perspectives, focusing on training-time data poisoning and inference-time prompt manipulation.

**Training-time data poisoning.** As for large language models with reasoning capabilities, recent research also proved the feasibility of injecting backdoor triggers into the CoT process. Jin et al. (2024) proposed SABER, which leveraged CodeBERT to find optimal positions for trigger insertion in the backdoor data curation process. Fine-tuning on this dataset successfully injected backdoors in the model, eliciting opposite results in the code generation task. Targeting the thinking length of reasoning models, BoT (Zhu et al., 2025c) embedded triggers to skip the thinking process, thereby affecting the answer quality. Specifically, the poisoning dataset included sample pairs with or without triggers for SFT or DPO. After that, Shadow-CoT (Zhao et al., 2025a) was also proposed to attack the internal reasoning, with a well-designed three-stage fine-tuning pipeline for backdoor injection without harming the general performance. Similarly, Chua et al. (2025) noticed the potential of the fine-tuning attack, and trained a "sleeper agent" to elicit bad behaviors only with trigger prompts, in which the CoT appeared either innocent or misaligned. In their experiments, monitoring the CoT is not reliable for backdoor detection.

**Inference-time prompt manipulation**. Inference-time prompt manipulation shares a huge overlap with prompt injection attacks (Yan et al., 2024; Clop & Teglia, 2024; Zhao et al., 2023), which *"aims to compromise the data of the target task such that the LLM-integrated application is misled to accomplish an arbitrary, attacker-chosen task"* (Liu et al., 2024b). Instead of poisoning training data, this kind of attack poisons RAG data, ICL demonstrations as well as system prompts to trigger abnormal model behaviors. Badchain (Xiang et al., 2024a) proposed to curate backdoor examples as demonstrations in ICL to elicit target generation. Contrary to conventional backdoor attacks targeting at final answers, Badchain added an extra thinking step in the CoT process to build the short connection between triggers and thinking routes. Moreover, evaluations in BackdoorLLM (Li et al., 2024g) further discovered that large language models with stronger reasoning capabilities are more vulnerable to backdoor attacks, a finding that mirrors the results in Jailbreak attacks (Wei et al., 2023a). Guo & Tourani (2025) proposed Darkmind, which altered model behaviors with modified instructions in the system prompt. After that, Guo & Cai (2025) tried multiple types of system prompts, finding that poisoned prompts with CoT or ICL could largely divert model outputs across various tasks. Under RAG settings, Song et al. (2025a) identified the ineffectiveness of simple knowledge editing, adding reasoning templates with erroneous knowledge into the system to camouflage reasoning models, which resembles the logic behind H-CoT (Kuo et al., 2025). In addition, Cui et al. (2025c) identified that inputting the thinking process with prompts into DeepSeek-R1 would prevent the model from generating a final answer, by which they designed a token-efficient prompt injection attack to trigger abnormal generation cessation and compressing the required number of tokens to about 2000 (Cui et al., 2025b). Following work by Cui & Zuo (2025) further reduced the required injection tokens to 109.

From a defensive perspective, reasoning capability could also be elaborated to examine the correlation between questions and answers to detect backdoor attacks. Li et al. (2024e) proposed Chain-of-Scrutiny (CoS) to analyze whether the model generation directly answers the prompts. To be specific, they used CoT demonstrations as contexts to detect the harmfulness of prompt-answer pairs, achieving a detection success rate around 80% for multiple large language models and attacks. Marinelli et al. (2025) proposed to identify prompt manipulations through the number of reasoning steps: if the prompt is injected with extra tasks, the step to follow instructions should be larger than expected. Similarly, Jin et al. (2025) proposed Guard, encompassing a judge agent and a repair agent for backdoored CoT detection as well as modification in code generation tasks. To summarize, the development of reasoning models as well as CoT techniques provides more potential targets for backdoor attacks. Except for outputting target harmful strings, new backdoor

attacks could force models to deviate from the proper thinking process, or directly interrupt the reasoning phase from fine-tuning or prompting, exposing higher risks of cutting-edge models than less capable models.

## 5 Robustness

According to Braiek & Khomh (2025), "*model robustness denotes the capacity of a model to sustain stable predictive performance in the face of variations and changes in the input data*". Robustness has always been a crucial part of trustworthy AI, as it determines whether a model can maintain stable and reliable performance when facing various adversarial noises in real-world deployments (Wang et al., 2022). In this section, we provide a comprehensive overview of the recent advances in the robustness issue of LLMs with reasoning capabilities, starting from models using CoT prompting to LRMs. Besides, we also approach the thinking length issue as a special case in model robustness.

### 5.1 Robustness Improvement with Reasoning Techniques

Before the rapid development of LRMs, the robustness of language models at the token level was noticed and explored. Xu et al. (2024d) found that providing a preemptive answer before reasoning contents could lead the model to generate a reasoning process that conforms to the given answer. Zhou et al. (2024a) added noisy rationales in in-context demonstrations, finding that large language models are hard to generate proper reasoning content, even with self-correction techniques (Huang et al., 2024a; Xi et al., 2023). Wang & Zhao (2024) proposed RUPbench to evaluate the reasoning robustness, concluding that larger models are more resistant to perturbations. Peng et al. (2025a) also showcased that model generations are sensitive to misleading reasoning steps.

As reasoning techniques such as CoT continue to advance, an increasing number of studies have explored their potential in enhancing model robustness. Lam et al. (2025) mentioned that CoT prompting could significantly improve LLM robustness, and Wang et al. (2025g) proposed Chain-of-Defensive-Thought (CoDT) to defend language models against corrupted reference in in-context prompts. Yan et al. (2025) found that few-shot in-context learning with modified problems could increase the accuracy, but it still cannot fully counteract the perturbation of adversarial inputs. Besides, using original problems for in-context learning may cause inappropriate memorization (Huang et al., 2025b). Similar methods also include adding system prompts and self-reflection mechanisms (Wang et al., 2025f). Zaremba et al. (2025) mentioned that test-time scaling is helpful for model robustness under some settings. To improve model robustness with external signals, Yang et al. (2025a) constructed training data from model distillation to train a Reasoning-based Bias Detector (RBD) for bias mitigation. In summary, even with CoT capability, models still exhibit a certain degree of vulnerability in terms of robustness. Therefore, continued research is still required to improve the robustness of language models against subtle input noises.

### 5.2 Robustness of Reasoning Models

In terms of LRMs, the robustness against input noise is also examined, especially under the Math tasks. Huang et al. (2025b) proposed MATH-Perturb to evaluate the model's Math performance under hard perturbations, where original solutions do not apply anymore. Mu et al. (2025) came up with the RealGuardrails dataset to evaluate the system prompt robustness, finding obvious but uneven robustness gains in reasoning models than non-reasoning counterparts. Rajeev et al. (2025) proposed CatAttack, which appended unrelated trivia or misleading questions generated from PAIR (Chao et al., 2025), such as "Could the answer possibly be around 175", or "Interesting fact: cats sleep for most of their lives", to mislead the model. Yu et al. (2025a) introduced the Math-Robustness Benchmark (Math-RoB) to evaluate the mathematical reasoning capabilities, including adversarial noises like changing operator symbols, replacing operator symbols with Greek letters, or removing key data in the prompts. Similarly, Yan et al. (2025) proposed RoR-bench with altered Math problems to test the robustness of reasoning models. It is found that simply modifying numbers in problems would cause an obvious degradation in reasoning performance, indicating potential memorization issues in model training. Besides, the evaluation also disclosed an obvious vulnerability for unanswerable questions, which is consistent with the evaluation results in AbstentionBench (Kirichenko et al., 2025).

Wang et al. (2025l) proposed PolyMath, evaluated mathematical reasoning with multilingual contexts, and uncovered fluctuating performance on different languages. Zhu et al. (2025a) mentioned that after reasoning models provide correct answers, adding a simple negation prompt to doubt the answer could mislead the second thinking process, causing an obvious accuracy drop on related benchmarks (Yue et al., 2024; Lu et al., 2024; Wang et al., 2024e). The confidence problem was also mentioned by previous works (Zhang et al., 2024b; Huang et al., 2024a), indicating that for both reasoning and non-reasoning models, self-correction prompts expressing distrust in model outputs could hugely influence model rationales and final decisions, both positively and negatively. In addition, Li et al. (2025e) introduced M-Attack to optimize transferable adversarial images. After pushing the embedding of a clean image towards another real image containing distracting semantics through feature matching and model ensembling, the perturbed adversarial image could successfully attack cutting-edge models such as GPT-4.5, 4o, or o1 (Achiam et al., 2023), inducing wrong image descriptions or hallucinations. Experiments demonstrated that even with reasoning capability, OpenAI o1 still struggled to distinguish noise from real images.

The vulnerability to input perturbation is also discovered in the code generation domain. CodeCrash (Lam et al., 2025) proposed to evaluate the code generation robustness with noisy requests, including garbage codes, renamed entities (which resembles altering numbers in Math problems), misleading print statements or hint comments, etc. While the results demonstrated superior performance compared to non-reasoning counterparts, they also revealed significant vulnerabilities under certain perturbations. Roh et al. (2025) identified the robustness vulnerability against the Chain-of-Code Collapse (CoCC) framework, in which the original prompt was wrapped with a narrative tone, making it a story or an adventure. Moreover, Wang et al. (2025f) evaluated the judging bias of large reasoning models, finding that even if LRMs perform better than LLMs on objective domains, they are still vulnerable to biases such as choice position, authority, or major beliefs distractions.

### 5.3 Overthinking and Underthinking

Overthinking is an emerging problem in reasoning models, referring to the phenomenon where "LLMs generate excessively detailed or unnecessarily elaborate reasoning steps, ultimately reducing their problem-solving efficiency". (Sui et al., 2025; Chen et al., 2024) From the trustworthy perspective, instead of efficiency, we focus more on situations where *models are trapped in repeating reasoning trajectories in a non-stop manner, and may output wrong answers in the end.* Conversely, underthinking refers to the situation where LLMs generate abnormally short reasoning or completely skip the reasoning process, even if the thinking behavior is necessary or required. Along the same lines as before, modifications to the Math questions could trigger redundant reflections, resulting in overthinking (Ma et al., 2024; Hashemi et al., 2025). Generally, such overthinking vulnerability mainly occurs when faced with unanswerable questions or erroneous premises. Some researchers (He et al., 2025; Fan et al., 2025a) found that the overconfidence, or reliance, on input prompts forces reasoning models to try numerous thoughts while failing to doubt the validity of prompts. Wang et al. (2025m) attributed the redundant thinking tokens with unsatisfying accuracy to frequent thought switching. Su et al. (2025) studied the relationship between reasoning length and answer correctness, finding that models failed to allocate proper reasoning length to questions with different levels of difficulty. Dang et al. (2025) also proposed that "internal bias" is strongly related to the overthinking behavior. When the internal bias contradicts the conclusion after stepwise thoughts, the model will trigger reflections.

To deliberately elicit overthinking behavior, the earliest work is Overthink (Kumar et al., 2025), which added unrelated or adversarial context to the prompts to obfuscate model reasoning. Similar attacks are also proposed in multiple literatures (Zaremba et al., 2025; Lam et al., 2025; Si et al., 2025), in which Si et al. (2025) introduced a GCG-style (Zou et al., 2023) optimization pipeline to generate adversarial overthinking triggers. Under agentic environments, Cuadron et al. (2025) identified the reasoning-action dilemma, and categorized three patterns of overthinking where the model prefers overly reasoning to interacting with environments. To mitigate overthinking, there are a lot of works heading towards efficient reasoning (Feng et al., 2025b; Qu et al., 2025; Sui et al., 2025), including but not limited to prompt-driven methods (Xu et al., 2025c; Ma et al., 2024; 2025b), training-based methods (Yang et al., 2025b; Wang et al., 2025e; Xia et al., 2025; Munkhbat et al., 2025; Yu et al., 2024a), inference-based methods (Huang et al., 2025a; Jiang et al.,

2025b; Wang et al., 2025k; Yu et al., 2025b), representation-based methods (Huang et al., 2025e; Cyberey & Evans, 2025), etc.

Underthinking, compared to overthinking, constitutes a more pure robustness topic. Input manipulation could also trigger underthinking (Cui et al., 2025c; Zaremba et al., 2025). For example, padding original prompts with compromised thoughts could make DeepSeek-R1 stop further reasoning (Cui et al., 2025c). A few researchers also mentioned that the think-less attack could limit the test-time compute of reasoning models, making them more vulnerable to attacks (Zaremba et al., 2025; Li et al., 2025c; Zhao et al., 2025b). Sun et al. (2025a) located a subset of attention layers in the model weight, proposed ThinkEdit to remove the short thinking direction. In general, current reasoning models lack sufficient robustness against manipulations of thinking length. To advance both robustness and efficiency, further research is needed to investigate the underlying causes of overthinking and underthinking behaviors, as well as to develop effective mitigation strategies.

# 6 Fairness

Fairness focuses on the ethical principles language models possess, especially whether language models react equally to different users or groups, including genders, LGBTQ+ communities, races, language, and political orientations without preference or discrimination (Huang et al., 2024b). As stated in previous literature (Li et al., 2023b; Gallegos et al., 2024), the bias may emerge, or be exaggerated, from imperfect training data, the choice of optimization, evaluation metrics, and the deployment phase. In this section, instead of thoroughly reviewing fairness evaluation and debiasing methods in LLMs, we simply limit our scope to recent fairness studies with regard to the reasoning capability.

Lin et al. (2025) identified the dialect bias of multiple cutting-edge language models with the experiments of paraphrasing standard English queries into African American Vernacular English (AAVE). CoT prompting is helpful to mitigate this bias, but it is unable to fully solve such a discrepancy, just like the results on robustness (Yan et al., 2025). Cheng et al. (2025d) also mentioned that CoT prompting could guide the model to correctly classify gender biases. Kamruzzaman & Kim (2024) evaluated multiple prompting strategies for social bias reduction, finding that system 2 prompts with a human persona could reduce stereotypical judgments. However, another line of work stated that under persona-assigned tasks, CoT prompts are not sufficient to mitigate human-like motivated reasoning (Dash et al., 2025; Gupta et al., 2024). For bias detection, Fan et al. (2025c) proposed BiasGuard to identify potential discrimination with internal reasoning capability. The training included an SFT stage followed by a DPO stage, which resembles the development of guardrail models in Section 4.2.3. Cantini et al. (2025a) exploited the CLEAR-Bias benchmark (Cantini et al., 2025b) for LRMs, concluding that models with explicit reasoning are more vulnerable in terms of bias, even though they are slightly safer than LLMs with CoT prompting. Overall, current researches underscore that current CoT and reasoning techniques have yet to bridge the gap toward achieving authentic fairness in models, and the fairness may still depend on the quality and distribution of training data.

# 7 Privacy

Privacy is always an important concern in the development of ML algorithms. Dating back to the CNN era, there has been a lot of work studying the potential to infer or steal the model and training data (Wang et al., 2024c; Shokri et al., 2017; Zhou et al., 2024b), as well as their corresponding defenses (Jiang et al., 2025c; Guan et al., 2022a; 2025). In recent years, we have also witnessed some inference-time attacks to extract personally identifiable information (PII), private retrieval-augmented generation (RAG) documents, or model weights when interacting with large language models (Jiang et al., 2024a; Wang et al., 2025n; Carlini et al., 2024). As reasoning capabilities become more advanced, the risk of intentionally disclosing private information through user input increases. In this section, we elaborate on related research from the model and prompt perspectives, specifically whether the privacy issue originates from model training data or external prompts.

## 7.1 Model-related Privacy

**Unlearning.** Large language model unlearning aims to erase copyrighted contents, remove harmful generations, protect data privacy, etc. (Yao et al., 2024b). Following previous work on unlearning method evaluation (Maini et al., 2024), Yoon *et al.* proposed R-TOFU (Yoon et al., 2025) to evaluate a few baseline unlearning methods with different strategies on reasoning models, concluding that unlearning only the final result is insufficient to forget the specific information. Similar conclusions are also drawn by Wang et al. (2025a), and they proposed $R^2MU$ that mapped the intermediate features of reasoning steps to randomly scaled vectors for an improvement. Both works highlighted the forgetting of CoT contents, providing a feasible direction for future attempts. From the other side, attacks against unlearning were also developed to recover erased data, which discloses the vulnerability of unlearning methods (Lynch et al., 2024; Hu et al., 2024). For reasoning models, Sinha et al. (2025) proposed SLEEK to elicit unlearned information in a multi-turn manner. Aimed at finding residual traces related to the unlearning target, SLEEK first generates queries targeting each object or fact with CoT techniques, and then prompts the model in multi-turn interactions to test whether any residual details remain in the response. This method achieved an ASR above 50% on Harry Potter facts against chat models, suggesting that full mitigation of memorized content may not yet be guaranteed.

**Model IP protection.** To prevent the model from copying or stealing, researchers have proposed numerous active or passive defense methods to protect the released models as well as their valuable training datasets, including fingerprinting, watermarking, unlearnable techniques, etc (Guan et al., 2022a; Peng et al., 2022; Wang & Chang, 2021; Guan et al., 2024b; Fernandez et al., 2023; Li et al., 2021; Fu et al., 2022; Sandoval-Segura et al., 2022; Huang et al., 2021). In terms of large language models, representing work (Kirchenbauer et al., 2023) promoted the sampling possibility of a fraction of tokens in the vocabulary, so that the watermark is printed as the ratio of selected tokens versus the rest tokens in the generated texts. After that, the development of CoT prompting provides more chances to model IP protection. ImF (Wanli et al., 2025) embedded the fingerprint[3] into pre-defined CoT prompt-answer pairs. CoTSRF (Ren et al., 2025) trained an extractor to capture the feature of CoT-prompt conditioned reasoning steps, and calculate the Kullback-Leibler divergence (KL divergence) with the suspect model in the verification phase. To enable RAG data protection, Guo et al. (2025c) imprinted watermarks into knowledge text, so that the model would generate a specific CoT trace with correct answers when faced with verification questions, enabling an effective and harmless copyright protection. Aside from watermarking methods, Savani et al. (2025) proposed "antidistillation sampling" to prevent model-generated contents from being trained. When decoding, the method modified the output logits to maximize the potential training loss while keeping the correctness of the outputs. Experiments on Math datasets (Hendrycks et al., 2021; Cobbe et al., 2021) demonstrated the feasibility of this approach: antidistillation sampling achieved accuracy comparable to temperature sampling, while student models suffered a notable performance drop of approximately 30% on GSM8K (Cobbe et al., 2021). Together, these techniques provide a basis for ongoing efforts to develop reliable and practical IP protection mechanisms.

## 7.2 Prompt-related Privacy

With the fast progress in large language models, the ability to infer private information from input prompts also gets stronger. Staab et al. (2024) was the first to research the privacy inference attack in large language models, drawing the result that LLMs are capable of inferring various personal attributes beyond memorization. Tömekçe et al. (2024) tested the inferring capability in the vision domain, demonstrated that the inference accuracy is positively related to the general capabilities of the models, and underscored the necessity of privacy protection methods. After the advent of CoT techniques, Green et al. (2025) evaluated the privacy leakage of reasoning models, claiming that the reasoning traces could disclose more private information. While additional reasoning steps may lead to more cautious final answers, they can inadvertently reveal sensitive data during intermediate generation, aligning with the findings discussed in Section 7.1 (Yoon et al., 2025; Wang et al., 2025a). Luo et al. (2025) curated a benchmark to evaluate the attribute inference attack of vision-language models, finding that multi-model large reasoning models have strong capabilities of

---

[3]"Fingerprint" originally refers to inherent, verifiable model features (e.g., weights or activations), while "watermark" denotes externally embedded signals. In this context, the distinction is blurred, and both terms refer to watermarks.

inferring geological information in input images, while seldom limiting this feature. Based on these findings, they proposed GeoMiner to trigger location-related attribute inference attacks. Such a method achieved higher performance than simple CoT methods, urging the need for protection.

With a similar logic to develop defense methods against Jailbreak in Section 4, the defense of attribute inference attacks also includes prompting, post-training, and guardrails. However, experiments by Staab et al. (2024) showed limited privacy gains from client-side anonymization or alignment. Such a vulnerability is also supported by Luo et al. (2025), stating that current SoTA guardrails cannot identify such an attack, and padding system prompts with warnings on location leakage could sacrifice the general performance. To summarize, more future works are needed to defend against this escalating threat.

# 8 Future Research Directions

**Standard measurements of faithfulness.** A wide range of methods have been proposed to evaluate reasoning faithfulness, but none are comprehensive, often leading to divergent or even contradictory conclusions. For example, some studies argue that larger models exhibit greater faithfulness (Bao et al., 2024; Xiong et al., 2025b), while others contend that they are less faithful (Bentham et al., 2024). This inconsistency highlights the need for more robust and standardized evaluation protocols that can fairly assess reasoning faithfulness across models.

In addition, some existing methods for evaluating faithfulness may conflict with other aspects of the performance of large models. For example, one common evaluation technique involves CoT intervention methods. These approaches test how perturbations to intermediate reasoning steps affect final answers. Empirical findings suggest that stronger models can answer correctly even with the perturbed CoT, implying that their outputs may rely less on explicit reasoning traces and more on internalized knowledge. From this, one might conclude that stronger models are less faithful, as their outputs do not depend transparently on the provided reasoning paths. However, such a conclusion conflicts with robustness. Therefore, eliminating the evaluation bias caused by model performance remains a critical open problem.

**More analyses on safety mechanism.** After reviewing attack and defense methods in Section 4, we call for more studies on the safety mechanism. Previous works demonstrated the feasibility of post-training methods with an extra safety-related CoT dataset. However, heuristic insights into effective dataset construction remain limited, leaving many details, such as prompts for CoT distillation, data ratios across different sources, and the necessity of cold-start SFT, reliant on manual tuning and empirical intuition. Moreover, in terms of the safety tax, the empirical understanding of how reinforcement learning contributes to safety and alignment remains limited. For instance, it remains challenging to disentangle the extent to which performance gains stem from the learning algorithm itself (e.g., GRPO over DPO) versus the influence of higher-quality data, such as well-curated CoT examples. Some progress has been made in understanding the role of SFT versus RL (Chu et al., 2025; Chen et al., 2025a), and we encourage future work to further investigate the role and limits of RL in this context.

**More fine-grained benchmarks**. As language models continue to grow in capability, there is an increasing need for safety evaluation benchmarks that can effectively reflect their evolving behaviors. Current safety evaluation benchmarks are primarily based on a narrow set of related attack methods (Zou et al., 2023; Röttger et al., 2024; Luo et al., 2024), resulting in significant homogenization of data distribution. As a consequence, metrics such as ASR often exhibit extreme values. Besides, due to the inherent properties of generative models, the outputs may be sensitive to variations in temperature settings and prompt formulations, thereby impacting the reproducibility of experimental results. In this regard, we call for new benchmarks that are more discriminative, detailed, and robust. In addition, compared with the number of benchmarks in safety and robustness, evaluations on privacy inference and fairness have comparatively received less emphasis. These areas would benefit from increased focus in future work if more evaluations with comprehensive coverage, clear definitions, and diverse testing samples are developed.

## 9 Conclusion

In conclusion, this survey summarizes recent literature concerning trustworthiness in reasoning capabilities, providing a comprehensive overview with a clear taxonomy. With efforts on each topic, we describe the development of novel methods, point out prevailing conclusions, and highlight the related analysis as well as future opportunities. We believe that our comprehensive survey and structured taxonomy could offer a foundation for future research in building safer, more reliable models with reasoning capabilities.

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
