# OpenReview forum: "A Comprehensive Survey on Trustworthiness in Reasoning with Large Language Models"
_TMLR — Rejected by TMLR_

### Review · Reviewer_ZVXB · 2025-09-29

**Summary Of Contributions:**

This work provides a systematic review of trustworthiness in large language model inference, covering aspects such as truthfulness, safety, robustness, fairness, and privacy. The breadth of literature surveyed is commendable, and the taxonomy-based structure contributes to a relatively clear organization. However, the review at times reads as a mechanical listing of prior work, with limited discussion on the broader implications or practical prospects of these approaches. Much of the manuscript introduces works strictly in chronological order. It would be more insightful if the authors could categorize methods based on their own perspectives and critical understanding of the field. The analysis of the tension between model size and trustworthiness is valuable, but the paper would benefit from deeper exploration, for instance by providing potential solutions or forward-looking insights. Overall, while the coverage and effort are notable, the paper currently resembles a catalog of methods, and its depth of analysis remains an area for improvement.

**Audience:**

Yes

**Audience Explanation:**

Given the growing attention to the safety, interpretability, and reliability of large language model reasoning, the topic of this survey is highly relevant to the readership of TMLR. From an objective perspective, a comprehensive investigation of this area has the potential to provide valuable guidance for both researchers and practitioners. However, the current manuscript primarily summarizes existing work without offering sufficient distinctive insights or critical reflections from the authors’ own perspective. Thus, while the paper is informative, there remains substantial room to strengthen its contribution.

**Broader Impact Concerns:**

This work does not appear to raise any significant ethical concerns.

**Claims And Evidence:**

Yes

**Claims Explanation:**

The claims made in this work are generally well supported by widely cited literature, yet the contribution remains somewhat limited. Much of the review tends to reiterate the conclusions of prior studies without thoroughly analyzing their potential shortcomings or the underlying causes of these issues. In other words, while the arguments presented are understandable, the paper lacks sufficient demonstration of the authors’ own distinctive insights.

**Requested Changes:**

1. As noted earlier, the survey’s approach to organizing prior work feels somewhat mechanical, which to some extent diminishes its contribution. I would encourage the authors to articulate their own perspective on the limitations of existing studies, the underlying causes of these limitations, and their potential implications for future applications.
2. The taxonomy of methods could be improved. Instead of classifying works merely from the perspective of the problems they address, I suggest exploring categorizations along dimensions such as methodological design principles or training strategies, rather than presenting them in a catalog-like manner.
3. The manuscript does highlight several important tensions in the field (e.g., model size versus trustworthiness, reasoning depth versus safety), but it does not attempt to propose potential resolutions. I recommend a more systematic discussion of the root causes of these conflicts, such as dataset biases, differences in evaluation protocols, or architectural variations.

---

> ### Comment · Action_Editor_QGP7 · 2025-11-19
>
> Dear Reviewer ZVXB,
>
> Would you send your office recommendation ASAP? Thank you!
>
>
>
> Best wishes,
>
>
> AE

---

> > ### Comment · Reviewer_5HCJ · 2025-11-19
> >
> > I submitted my official recommendation on 28 October.

---

### Review · Reviewer_5HCJ · 2025-10-10

**Summary Of Contributions:**

The title of this paper is nicely informative: it is a survey paper
where the emphasis is on being comprehensive. The topic surveyed is
that of trustworthiness in reasoning in LLMs. The writing is clear and
with very few grammatical errors.

As far as I can judge the submission is indeed as comprehensive as one
can hope for given the fast rate of publication in this area. There
are very many papers mentioned (~300) and they are relevant to the
paper's topic. It is also notable that many (I think most) of the
papers referenced are from 2025, so the paper considers the latest
research. I think the taxonomy in Figure 2 is very useful, allowing a
reader to readily relate the various sub-topics and find the section
of the survey where that sub-topic is examined. (The list of acronyms
on Table 1 is also helpful.) I should add that I do not work in this
area, so am not in a position to spot relevant papers that have been
missed.

The key question is whether this survey paper is suitable for
TMLR. For a survey paper, TMLR wants "surveys that draw new
connections, highlight trends, and suggest new problems in an area."
Although I think it fair to say that the survey "highlights trends" I
think it falls short on making new connections and suggesting new
problems. The (entirely reasonable and indeed achieved) goal of the
paper is a synthesis (of results and insights in the literature) to
provide a resource to interested readers. I think for TMLR, at least,
we need something more (or rather different) with a greater emphasis
on novel insights coming from the survey.

On this key criterion, consider the following text from the paper:
"Through our review of existing work, we suggest that reasoning
techniques not only facilitate the development of more interpretable
and trustworthy models but also introduce new vulnerabilities." But
this suggestion does not really emerge from the review per se, it is a
point made by very many of the papers reviewed.

As far as pointers to future work, suggestions are often made at the
end of sections. For example on page 8 we have a typical example:
"Addressing this tension between reasoning capability and reliability
will require systematic investigation, and stands as an important
direction for future research." This is a true statement but merely
noting the need for some sort of "systematic investigation" is weak;
posing a specific research question would be much better.

MINOR POINTS:

What does "FUR" mean?

p16: "Except for alignment methods that would be covered in detail in
Section 4.3," should be "Except for alignment methods that are
covered in detail in Section 4.3,"

Fig 3 should be referenced in the text.

Occasionally, the present tense is used where the past tense would
read better. For example on p21 we have: "It is found that simply
modifying numbers". "It was ..." is better.

p25: "However, such a conclusion conflicts with robustness." I did not
understand this.

**Audience:**

Yes

**Audience Explanation:**

For someone wanting an overview of recent work in this area, nicely organised into sub-topics, this is a useful resource.

**Broader Impact Concerns:**

The paper surveys an area where ethical issues abound. But I do not think this paper requires eg a "Broader Impact Statement".

**Claims And Evidence:**

Yes

**Claims Explanation:**

The paper is a survey paper which only make weak claims (that some area of the topic surveyed requires further research). The unresolved nature of many research questions in this area are sufficient to support such weak claims.

**Requested Changes:**

To meet the TMLR criteria for survey papers we need more analysis, critique and specific suggestions for future research. A comprehensive synthesis, useful as it is, does meet these criteria. This would of course be a major revision and is a lot more than dealing with a "list of proposed adjustments"

---

### Review · Reviewer_4rtd · 2025-10-13

**Summary Of Contributions:**

This survey paper provides a comprehensive overview of trustworthiness issues in reasoning-capable large language models (LLMs), focusing on both Chain-of-Thought (CoT) techniques and Large Reasoning Models. The authors organize recent literature across five core dimensions: truthfulness (hallucination and faithfulness), safety (vulnerability, jailbreak, alignment, backdoor), robustness (adversarial inputs, overthinking/underthinking), fairness (bias detection), and privacy (model and prompt-related), which positions trustworthy reasoning as a distinct research focus, examining both reasoning as a tool for improving trust (e.g., self-reflection, consistency checks) and reasoning as a new attack surface (e.g., reasoning-based jailbreaks, over-thinking). The authors then propose a taxonomy, synthesize representative studies between 2022 and 2025, and highlight key insights, such as the tension between faithfulness and accuracy, multilingual safety gaps, and emerging reasoning-specific failure modes (over/under-thinking).

In brief, the reviewer finds the submission helpful in terms of:

1. It studies a timely and well-defined topic, focusing specifically on reasoning trustworthiness of LLM safety.
2. It presents a clear five-pillar taxonomy that organizes the literature. This organizational framework is described with well-defined trustworthiness dimensions.
3. It includes a helpful background section that introduces CoT and large reasoning models related concepts.

Moreover, the taxonomy that visualizes relationship between topics is helpful; the breadth of this survey covering literature across multiple safety aspects is commendable.

However, the reviewer finds that there exist several limitations that the authors are expected to resolve: (to keep this section focused, here only the opinions are included; please see the next section (Explain your answer above) where the reviewer's opinion is justified.

1. This submission predominantly adopts a catalog approach, which lists what papers exist and briefly describing what each does. It however does not provide analytical synthesis that helps to reveal underlying patterns, principles, or insights.
2. This submission rarely compares different methods' effectiveness, namely it does not evaluate strengths/weaknesses of competing approaches, and provides minimal guidance on when to use which technique.
3. This submission exhibits disparity in coverage across trustworthiness dimensions.
4. This survey rarely aggregates findings across studies, nor provides quantitative synthesis, and provides no meta-analytic conclusions about what works and why.
5. When related studies reach opposite conclusions, this survey simply reports both without analyzing (or attempting to) why they disagree.

**Audience:**

Yes

**Audience Explanation:**

This paper addresses a highly relevant and timely topic that would interest substantial segments of TMLR's audience. The topic has practical importance. As reasoning models like OpenAI's o1, DeepSeek-R1, and others are rapidly deployed in real-world applications, understanding their trustworthiness challenges is urgent for researchers and practitioners. The volume of recent work demonstrates strong community interest in this topic. In addition, the transition from standard LLMs to reasoning-capable models represents a significant paradigm shift with implications for how we think about model capabilities, vulnerabilities, and safety mechanisms. Lastly, the identified research gaps and future directions (though currently underdeveloped) point to open problems that would interest TMLR's research-oriented audience.

However, the paper should undergo substantial revision to realize this potential value by transforming from a literature list into a critical synthesis that provides deeper insights and guidance for the community.

**Broader Impact Concerns:**

No major ethical concerns, but recommend adding a brief subsection to discuss the dual-use nature of vulnerability research.

**Claims And Evidence:**

No

**Claims Explanation:**

1. For W1,
    - in section 4.1 for example, the submission enumerates Jiang et al. (2025b), Zhou et al. (2025a), Kassianik & Karbasi (2025). However, key insights and questions, such as why do these models show such different safety profiles? what architectural or training differences explain the 100% ASR vs. better performance? what pattterns emerge across these evaluations? what principles should guide future model development? are omitted from the paper.
    - In section 3.1.2, the paper lists "Several studies (Dong et al., 2025a; Song et al., 2025b; Liu et al., 2025a; Yao et al., 2025b; Cheng et al., 2025a; Kirichenko et al., 2025) have documented significant hallucination issues within reasoning models..." but it fails to quantify the magnitude, identify which types of hallucinations increase most, and compare the relative importance of different contributing factors. In this sense, readers finish sections knowing "literature x, y, z exist" but lack understanding of what the field has collectively learned, which approaches are most promising, what fundamental trade-offs exist, where critical gaps remain, and (potentially) how to make informed methodological choices.

2. For W2
    - in section 3.2.3, the survey lists six different approaches, namely Question decomposition, Faithful CoT with symbolic reasoning, LOGIC-LM, SymbCoT, FLARE, and CoMAT. But it does not show which methods work best on which task types, computational costs (since symbolic solvers may be slower), robustness to different perturbations, external resource requirements, etc., which is preferably to be explicitly discussed and compared with a comparative table.
    - in section 4.2.3, the survey lists eleven different guardrails models but provides no comparison of detection accuracy and computational costs, no discussion of false positive/negative rates, also no guidance on selection criteria. It means there are a number of key questions left unanswered, e.g., is GuardReasoner better than RSafe, under what conditions? Do GRPO-based methods outperform SFT-only approaches? Which methods generalize better to new attack types?
    - in section 4.3.1, the survey describes ten different alignment methods but does not compares relative safety improvements and safety tax, as well as robustness to different attack types.

3. For W3, the paper spent different page to elaborate on different dimensions (i.e., truthfulness, safety, robustness, fairness, and privacy). This makes the "trustworthiness" assessment incomplete - a system could be safe but unfair, or private but biased. The current structure implies trustworthiness $\approx$ safety + truthfulness, which is inadequate.
    - the fairness section contains no discussion of bias types, sources, and evaluation methodologies.
    - the privacy section still lacks formal privacy definition and comprehensive threat models. The reviewer also reckons that privacy-preserving reasoning and regulatory compliance considerations should be mentioned.
    - there also lacks sections that discuss the connections between dimensions, e.g., can safety alignment introduce or amplify biases? do privacy-preserving techniques affect fairness?

4. For W4, the paper refers several evaluations, e.g., Kassianik & Karbasi (2025) reported that ASR of DeepSeek-R1 on Harmbench is 100%, Zhou et al. (2025a) showed that o3-mini significantly safer than DeepSeek-R1, and Krishna et al. (2025) studied category-wise vulnerability. But it is expected to have a meta-analysis table that summarizes metrics like ASR for open/closed-source reasoning models across studies. On the other hand, this paper lacks cross-study comparisons, multiple methods are described but are not compared compared quantitatively.

5. For W5
    - for example, section 3.2.2 the paper states that Lanham et al. (2023), Bentham et al. (2024) and  Bao et al. (2024), Xiong et al. (2025b) argued contradict findings regarding the model size and faithfulness. Section 4.1 also lists the converse finding regarding reasoning impact on safety. Section 3.1.1 and 3.1.2 cover the contradiction between Cheng et al. 2025a and LRMs exacerbate hallucination issues...higher rates of hallucination.

**Requested Changes:**

1. For each major subsection (especially sections 3.1, 3.2, 4.1, 4.2, 5.2), add comparative analysis identifying what makes approaches effective/ineffective, common principles underlying successful methods, resolution of contradictory findings with analysis of why discrepancies exist.
2. Include tables comparing hallucination detection methods (metrics, effectiveness, computational cost), jailbreak defense mechanisms (approach type, ASR reduction, computational overhead), faithfulness measurement techniques (intervention type, strengths, limitations), safety alignment methods (training approach, data requirements, safety-performance trade-off).
3. Expand Fairness and Privacy sections, covering (for fairness) bias sources, evaluation methodologies, mitigation strategies, fairness-performance trade-offs; (for privacy) formal definitions and privacy-utility trade-offs.

---

### Decision · Action_Editor_QGP7 · 2025-12-22

**Recommendation:** Reject

**Additional Comments:**

In general, I appreciate the author's effort in writing a comprehensive survey paper. On the other hand, some of the concerns raised by the reviewers are not sufficiently addressed. I would gently recommend that the author fix those issues to enable the paper to "draw new connections, highlight trends, and suggest new problems" as a survey paper, and consider resubmission in the future volume of TMLR.

**Audience:**

Yes

**Audience Explanation:**

This is a timely survey on an important topic; the ML community would generally be interested in it.

**Claims And Evidence:**

No

**Claims Explanation:**

The statements with regard to the analysis of limitations or potential solutions for the mentioned challenges are not very convincing, with clear evidence.

Additionally, as a survey paper, we would expect it to "draw new connections, highlight trends, and suggest new problems" if accepted by TMLR. The current version of the manuscript does not sufficiently reach this goal.

**Resubmission Of Major Revision:**

The authors may consider submitting a major revision at a later time.